# DOES DATASET LOTTERY TICKET HYPOTHESIS EXIST?

## ABSTRACT

Tuning hyperparameters and exploring the suitable training schemes for the self-supervised models are usually expensive and resource-consuming, especially on large-scale datasets like ImageNet-1K. Critically, this means only a few establishments (e.g., Google, Meta, etc.) have the ability to afford the heavy experiments on this task, which seriously hinders more engagement and better development in this area. An ideal situation is that there exists a subset from the full large-scale dataset, the subset can correctly reflect the performance distinction[1] when performing different training frameworks, hyper-parameters, etc. This new training manner can substantially decrease resource requirements and improve the computational performance of ablations without compromising accuracy using subset discovered configuration to the full dataset. We formulate this interesting problem as the *dataset lottery ticket hypothesis* and the target subsets as the *winning tickets*.

In this work, we analyze this problem through finding out partial empirical data on the class dimension that has a consistent *Empirical Risk Trend* as the full observed dataset. We also examine multiple solutions, including (i) a uniform selection scheme that has been widely used in literature; (ii) subsets by involving prior knowledge, for instance, using the sorted per-class performance of the strong supervised model to identify the desired subset, WordNet Tree on hierarchical semantic classes, etc., for generating the target winning tickets.

We verify this hypothesis on the self-supervised learning task across a variety of recent mainstream methods, such as MAE, DINO, MoCo-V1/V2, etc., with different backbones like ResNet and Vision Transformers. The supervised classification task is also examined as an extension. We conduct extensive experiments for training more than **2K** self-supervised models on the large-scale ImageNet-1K and its subsets by **1.5M** GPU hours, to scrupulously deliver our discoveries and demonstrate our conclusions. According to our experimental results, the winning tickets (subsets) that we find behave consistently to the original dataset, which generally can benefit many experimental studies and ablations, saving $10\times$ of training time and resources for the hyperparameter tuning and other ablation studies.

## 1 INTRODUCTION

In the recent years, large deep neural networks, such as Convolutional Neural Networks (CNNs) (Lecun & Bengio, 1995; He et al., 2016) and Transformers (Vaswani et al., 2017) have achieved breakthroughs in the fields of supervised learning (Tan & Le, 2019; Dosovitskiy et al., 2020) and self-supervised learning (Kenton & Toutanova, 2019; Brown et al., 2020; Caron et al., 2021; 2020; He et al., 2022) empowered by the large-scale datasets. Naturally, the computational resources required for training these models on the large data are increasing accordingly. A dilemma is that many researchers do not have such resources to conduct experiments on the large datasets directly, especially on the expensive self-supervised learning by tuning the hyperparameters and exploring the proper training settings and frameworks. A commonly-used practice in the vision domain is to conduct ablations on relatively smaller datasets like CIFAR (Krizhevsky, 2009) and MNIST (Lecun et al., 1998), and then transfer the tuned optimal configurations to the large datasets like ImageNet-1K (Deng et al., 2009). While in many cases, it is observed that the models' behaviors and properties

---

[1]In this work, we focus on the *performance trend* or *relative accuracy trend* trained on the subset and full data across different train/eval configurations. The **absolute** accuracy on the individual subset is not necessary.

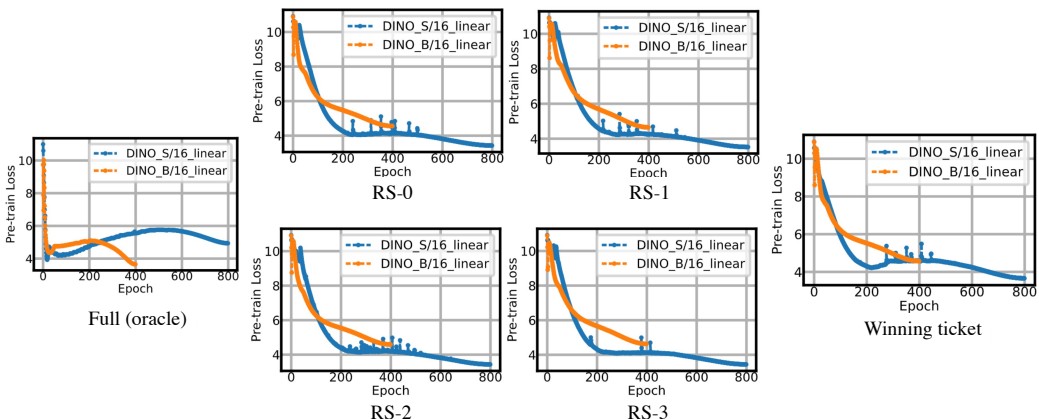

Figure 1: Illustration of the loss curves from self-supervised pre-training on full data, random subsets and winning ticket subset. ViT-Small and ViT-Base models are used as the backbone networks.

on the small datasets are quite different from the ones learned on the large dataset, making it improper to directly transfer hyperparameters found on small data to the large one. Considering that self-supervised learning does not use human-annotated labels for training models, another popular solution emerged by randomly choosing a subset from the full dataset, for example, randomly selecting 100 classes among 1,000 in ImageNet-1K for self-supervised ablations and exploring experiments (Tian et al., 2020; Kalantidis et al., 2020; Ermolov et al., 2021). Such a scheme has shown great advantages in lower resource demand of costly learning frameworks for fast hyperparameter tuning and model exploration with large backbone architectures.

However, according to the learning rule of Empirical Risk Minimization (ERM) (Vapnik, 1991; 1999) principle, the training convergence of ERM is guaranteed when the number of parameters of the neural networks scales linearly with the number of training examples. Under this principle, it is challenging to only leverage a subset of data to model the properties of the full larger number of training data, since in both of the settings, models are trained to minimize their average error over the current training samples. Moreover, ERM is unable to provide generalization on unseen distributions from the subset to the whole, making this strategy full of uncertainty. Consequently, a natural concern has been raised in this work: *What kind of subset is qualified for evaluating self-supervised/supervised methods on full data?*

**Goal of Dataset Lottery Ticket Hypothesis (DLTH):** The goal of DLTH is to find out a subset as the winning ticket from a large-scale dataset, this subset has the same or similar empirical behaviors and performance trends as the original full dataset when performing different training approaches and hyper-parameters on it, meaning that it can truly reflect the performance changes according to different training settings. Different from **(i)** data pruning (Zhang et al., 2021; Sorscher et al., 2022) that removes low-contribution and forgetting data to replace the full dataset, **(ii)** dataset distillation (Wang et al., 2018; Cazenavette et al., 2022) and condensation (Zhao et al., 2020) that will generate a new compressed data, the proposed DLTH will not predict the accuracy of full data but select a proper subset from the original data and the models trained on the subset under various configurations have a consistent performance trend to the models trained on the full data. Thus, we can further use this subset for fast hyper-parameter tuning, frameworks exploration or time-consuming tasks. More detailed discussions with the related tasks are provided in Appendix G.

**Overfitting Issue on the Subset:** The key observed issue of using subset data is that the self-supervised pre-training on the smaller training data (subset) will frailly suffer from overfitting. To reveal this, we visualize the pre-training losses in Fig. 1. Following DINO protocol (Caron et al., 2021), we train with 400 epochs for ViT-Base and 800 epochs for ViT-Small. On the full data, the evolution of training loss first drops rapidly, then slowly rises a little bit, and finally continues to descend on both small and base models. While on the randomly selected subsets (RS-ID), the losses of base and small models are dropping constantly with a plateau. It is interesting to see on our identified winning ticket, the rebounding phenomenon on loss has emerged again at around 400 epoch of blue curve and the magnitude of loss value is generally larger than those on the random subsets, which is also more aligned to the trend of the full data.

**Identifying Winning Tickets (Subsets).** The most common practice is to randomly select sub classes among the full data's categories as the subset (Tian et al., 2020), however, it is natural that a randomly selected subset is not guaranteed for reflecting the true accuracy change on the full data when imposing different training settings. To this end, we propose to fix the learned hypothesis or models on the full observed dataset, and identify the winning ticket which has the approximating distribution as the full observed data empirical distribution. We argue that the subset categories should have the consistency on performance similar to that of the full dataset. For instance, the overall linear probing performance of DINO (Caron et al., 2021) is better than MAE (He et al., 2022) with the same ViT-B/16 backbone, while on the specific class such as *toy poodle*, MAE has the better accuracy than DINO, meaning that this class has an inconsistency on the two self-supervised models and will not be selected. The consistency indicates that the individual per-class accuracy should match the global accuracy across different frameworks[2]. We also empirically examine other policies on determining the proper winning ticket subsets, including: **(i)** randomly generated lists as the baselines; **(ii)** incorporating prior knowledge (e.g., performance-driven scheme according to the pre-trained supervised models, semantic hierarchy on WordNet Tree, etc.).

The practice of randomly selecting a subset to evaluate different methods is more popular in self-supervised learning task. Thus, it is more crucial to clarify and understand this manner in self-supervised domain. While, supervised learning is also eligible to explore the dataset lottery ticket hypothesis and we provide the results on supervised learning in Appendix due to the limited space. All of our models are trained on large-scale ImageNet-1K and its subsets instead of MNIST, CIFAR, etc., with 3 runs for each, to avoid potential mis-observations and deliver more reliable conclusions.

**Contributions:**

- Our experimental results indicate that a randomly selected subset without any prior knowledge is unstable and generally not qualified for reflecting the properties of self-supervised models on the full data, and might further deliver misleading observations and conclusions. This is fairly risky if the studies rely heavily on such a kind of subset, instead of the original full dataset.

- We propose the Dataset Lottery Ticket Hypothesis (**DLTH**), a novel problem that studies the possibility of identifying the subset which can reflect the performance consistency with the full data. Moreover, we propose the policies of *Empirical Risk Trend* and *incorporating prior knowledge* to generate the winning tickets. To our best knowledge, this is the first work to study the feasibility of dataset lottery ticket hypothesis, and it can be a starting point for exploration on this problem.

- We provide comprehensive experiments on ImageNet-1K and its subsets with a variety of advanced self-supervised frameworks, such as DINO (Caron et al., 2021), MAE (He et al., 2022), MoCo series (He et al., 2020; Chen et al., 2020b). to verify the effectiveness and superiority of our proposed dataset winning ticket policies. We will make our found winning ticket subset publicly available to benefit other research in this field.

## 2 FROM EMPIRICAL RISK MINIMIZATION TO DATASET LOTTERY TICKET HYPOTHESIS

In the learning problem, a model is used to learn a conditional probability distribution or decision function. The hypothetical or mapping space of the model contains all the conditional probability distributions or decision functions. The goal of learning is to find an ideal hypothesis or mapping function $h \in \mathcal{H}$ with its parameter $\theta$ by observing the properties of the hypothesis space. Assume the input $X$ and target $Y$ of the model are random variables, which follow the joint probability distribution $P(X, Y)$, we define a loss function $\mathcal{L}$ that measures the differences between predictions $h(x)$ and corresponding targets $y$, i.e.,$(x, y) \sim P$. Usually, the smaller the value of the loss function $\mathcal{L}$ on validation set, the better the model learned by the optimization. The expectation of the loss function is formulated as:

$$R_{\exp}(h) = \mathbf{E}_P[\mathcal{L}(h(x; \theta), y)] = \int_{X \times Y} \mathcal{L}(h(x; \theta), y) P(x, y) dx dy \quad (1)$$

The learning perspective usually treats the optimization problem as simply minimizing the expected loss or called risk function. The distribution $P$ is unknown in most practical situations. Instead, we

---

[2]As shown in Fig. 3, on the linear probing evaluation, it is intriguing to see that DINO has the better overall accuracy, while MAE is more stable on individual accuracy across different categories.

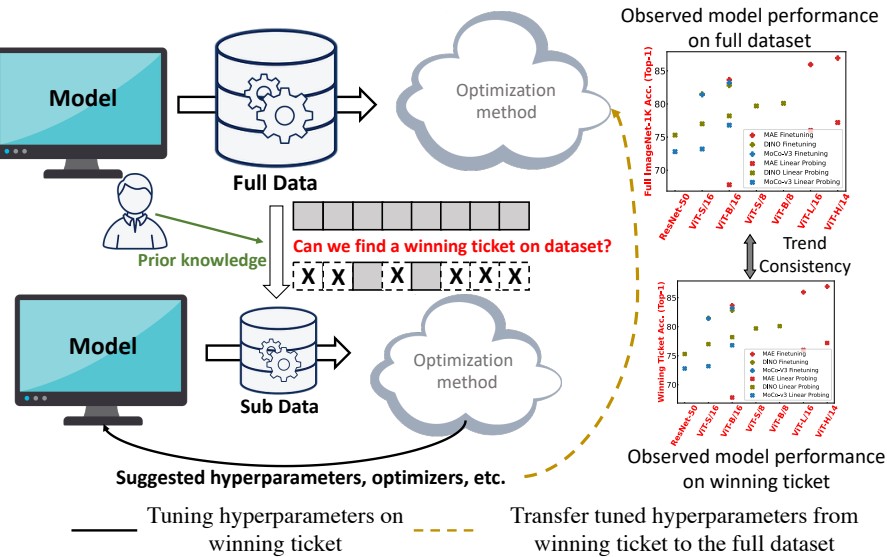

Figure 2: Illustration of the motivation on the proposed DLTH. We aim to identify a subset from the original data through leveraging prior knowledge as the dataset winning ticket, meanwhile, the subset should have the property that the observed model performance on it is consistent with the performance on full data across different training frameworks, settings and hyperparameter choices.

have access to a set of training data $\mathcal{D} = \{(\boldsymbol{x}_i, \boldsymbol{y}_i)\}$ $(i = 1, 2, \ldots, m)$, we can leverage the empirical distribution $\hat{\boldsymbol{P}}(\boldsymbol{X}, \boldsymbol{Y})$ on the available training data $\mathcal{D}$ to approximate the true distribution $\boldsymbol{P}$. Thus, we further can approximate the expected risk according to the empirical risk:

$$R_{\text{emp}}(\boldsymbol{h}) = \frac{1}{\boldsymbol{m}} \sum_{i=1}^{\boldsymbol{m}} \mathcal{L}\left(\boldsymbol{h}\left(\boldsymbol{x}_i; \boldsymbol{\theta}\right), \boldsymbol{y}_i\right) \quad (2)$$

Through a proper optimizer, a set of parameters is found on the training set so that the function can approximate the real mapping relationship. The next step is to learn such a function $\boldsymbol{h}^*$ (i.e., finding the learnable parameters $\boldsymbol{\theta}^*$) that the empirical risk (ER) is minimal:

$$\boldsymbol{h}^* = \underset{\boldsymbol{h} \in \mathcal{H}}{\arg\min} \, R_{\text{emp}}(\boldsymbol{h}) \quad (3)$$

According to the learning process of empirical risk, in this paper, we study the problem of whether there exist smaller available subsets as the winning tickets for training self-supervised models with similar performance trends to the full data. Training on the subsets is faster since the data size is significantly smaller than of the original data. Fig. 2 shows the motivation that we aim to identify a winning ticket on the dataset dimension. Here, we formally state the *dataset lottery ticket hypothesis*.

**Definition 1** (Dataset Lottery Ticket Hypothesis). *A large-scale, naturally-collected dataset contains a subset that has the same or similar empirical behaviors and performance trends as the original full dataset when performing different training approaches and hyper-parameters on it.*

Formally, in our *Dataset Lottery Ticket Hypothesis* setting, we have the learned risk function $\boldsymbol{h}^*$ (such as MAE (He et al., 2022), DINO (Caron et al., 2021), SwAV (Caron et al., 2020), etc.) together with their parameters $\boldsymbol{\theta}^*$ (trained models) from the full training data, our goal is to identify a subset of training data $\mathcal{D}_s = \{(\boldsymbol{x}_i, \boldsymbol{y}_i)\}_{i \in \mathcal{T}}$ as a winning ticket that the ER is minimal and consistent to different risk functions. We can leverage the subset data's empirical distribution $\hat{\boldsymbol{P}}_s(\boldsymbol{X}, \boldsymbol{Y})$ as the available empirical risk:

$$R_{\text{emp}}^{\mathcal{D}_s}(\boldsymbol{h}) = \frac{1}{|\mathcal{T}|} \sum_{i \in \mathcal{T}} \mathcal{L}\left(\boldsymbol{h}^*\left(\boldsymbol{x}_i; \boldsymbol{\theta}^*\right), \boldsymbol{y}_i\right) \quad (4)$$

Where $\mathcal{T}$ is the sample index of dataset winning ticket. Different from the usual learning goal that is to find an optimal function $\boldsymbol{h}^*$, identifying the winning ticket on the dataset is an *ill-formed* problem

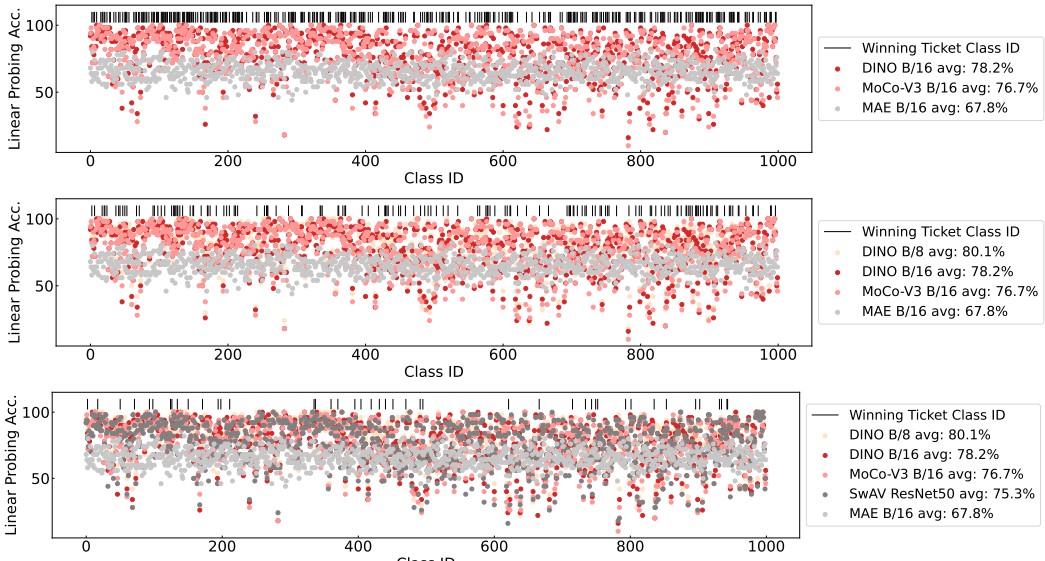

Figure 3: Illustration of per-class Top-1 accuracy of different trained models on full ImageNet-1K validation set. In each subfigure, the black lines above the bubbles are the winning classes. From top to bottom subfigures, there are 367, 155 and 44 classes with particular distribution density that meet the conditions of accuracy trend on different frameworks. The specific names of targeted class IDs are provided in Appendix. More details are in Sec. 3.1.

with uncertainty and randomness, our idea is to make it inevitable by considering that the individual selected class (ticket) should have the consistent performance behaviors as it on the full data. Note that the pre-trained parameters $\boldsymbol{\theta}^*$ are always fixed during this procedure. Thus, we identify the targeting class according to the consistency on the individual class empirical risk, we can find out the winning classes by maximizing the consistency metric which is defined as follows:

$$\mathcal{D}_s^* = \underset{\mathcal{D}_s \in \mathcal{D}}{\arg\max}\, \mathcal{C}_{\text{cons}}([R_{\text{emp}}^{\mathcal{D}_s}(\boldsymbol{h}_j^*)]_{j \in \mathcal{P}}) \tag{5}$$

where $\mathcal{C}_{\text{cons}}$ is the consistency policies for discovering dataset winning tickets. $\mathcal{P}$ is the set of hypothesis functions, i.e., the learning methods. Eq. 5 indicates that when given the learned models (hypothesis functions and their parameters) on the observed full dataset $\mathcal{D}$, the dataset lottery ticket hypothesis is to find a subset $\mathcal{D}_s$ as the winning ticket that meets the consistency relationship among different hypothesis functions. So that we can use this subset for different usages, such as fast tuning hyperparameters. We also examined the generalizability of the winning ticket for assessing new learning functions.

After acquiring the desired dataset winning ticket (subset), we can tune the hyperparameters and re-train the model following:

$$\min_{\boldsymbol{\theta}} \mathcal{L}(\boldsymbol{\theta}, \mathcal{D}_s^*) = \min_{\boldsymbol{\theta}} \sum_{(\boldsymbol{x}, \boldsymbol{y}) \in \mathcal{D}_s^*} \ell\left(\boldsymbol{h}(\boldsymbol{x}, \boldsymbol{\theta}), \boldsymbol{y}\right) \tag{6}$$

We also provide a brief explanation from the theory perspective in Appendix F.1.

## 3 WINNING TICKETS POLICIES

### 3.1 PER-CLASS EMPIRICAL RISK CONSISTENCY

In this section, we introduce the policy of identifying the winning ticket by leveraging the per-class empirical risk consistency.

---

**Algorithm 1** Per-class Empirical Consistency

**Input:** $\{\boldsymbol{h}_j(j \in \mathcal{P})\}$ is the pre-trained model pool with linear probing classifiers to identify the desired classes based on the consistency of per-class empirical accuracy, $\boldsymbol{C}$ is the number of classes in the dataset, $\mathcal{S}_c$ is the validation samples in class $\boldsymbol{c}$. $\boldsymbol{T}$ is the threshold if the identified size of winning ticket is too large. $M_{\text{target}}$ is the desired size of winning ticket after a random selection operation.

**Output:** Selected winning ticket $\mu^*$

1: **for** $\boldsymbol{c} = 1 : C$ **do**
2:   **if** $\mathcal{C}_{\text{cons}}^c([\sum_{i \in \mathcal{S}_c} R_{\text{emp}}(\boldsymbol{h}_j(\boldsymbol{x}_i, \boldsymbol{\theta}))]_{j \in \mathcal{P}})$ is *True* **then**
3:     $\mu.\text{append}(\boldsymbol{c})$
4:   **end if**
5: **end for**
6: **if** $\text{len}(\mu) > \boldsymbol{T}$ **then**
7:   $\mu^* = \text{Random\_Selection}(\mu, M_{\text{target}})$
8: **end if**
9: **return** $\mu^*$

---

The procedure is as follows:

**(i)** Train standard self-supervised backbone models $\mathcal{P}$ on the original full data.

**(ii)** Train supervised linear probing classifiers or finetune above trained backbones on the full dataset.

**(iii)** Calculate per-class prediction using the above classifiers and test the models on the validation set to match the result on the full data across frameworks. This is to identify that the particular class has the same accuracy trend as that on the full data, i.e., consistency condition between per-class accuracy and global accuracy on different models.

Specifically, there exists a well learned model/hypothesis pool $\mathcal{P}$, the criterion of consistency follows their empirical risk on the full dataset. For instance, if we consider three self-supervised methods: DINO, MAE and MoCo V3 with ViT-B/16 as the backbone network under linear probing evaluation, the consistency metric $\mathcal{C}_{\text{cons}}$ will be *True* only if it meets the following condition (the higher of the model accuracy, the lower $R_{\text{emp}}$):

$$\mathcal{C}_{\text{cons}} = [R_{\text{emp}}(\boldsymbol{h}_{\text{DINO}_{\text{ViT-B/16}}}) < R_{\text{emp}}(\boldsymbol{h}_{\text{MoCo V3}_{\text{ViT-B/16}}}) < R_{\text{emp}}(\boldsymbol{h}_{\text{MAE}_{\text{ViT-B/16}}})] \tag{7}$$

For class $\boldsymbol{c}$, if $\mathcal{C}_{\text{cons}}^{\boldsymbol{c}}$ is *True*, we will select it as one class of the winning ticket. We formulate it as:

$$\mathcal{C}_{\text{cons}}^{\boldsymbol{c}} = [\sum_{i \in \mathcal{S}_{\boldsymbol{c}}} R_{\text{emp}}(\boldsymbol{h}_{\text{DINO}_{\text{ViT-B/16}}}(\boldsymbol{x}_i)) < \sum_{i \in \mathcal{S}_{\boldsymbol{c}}} R_{\text{emp}}(\boldsymbol{h}_{\text{MoCo V3}_{\text{ViT-B/16}}}(\boldsymbol{x}_i)) < \sum_{i \in \mathcal{S}_{\boldsymbol{c}}} R_{\text{emp}}(\boldsymbol{h}_{\text{MAE}_{\text{ViT-B/16}}}(\boldsymbol{x}_i))] \tag{8}$$

The results on linear probing eval of the per-class empirical risk consistency is shown in Fig. 3. In practice, we use per-class validation accuracy as the consistency indicator. We examine three pre-trained model pools: (1) $R_{\text{DINO}_{\text{ViT-B/16}}} - R_{\text{MoCo V3}_{\text{ViT-B/16}}} - R_{\text{MAE}_{\text{ViT-B/16}}}$ as in Fig. 3 (upper sub-figure). The black lines above the bubbles are the targeted winning classes, it is observed that 367 classes meet the requirement in ImageNet-1K; (2) $R_{\text{DINO}_{\text{ViT-B/8}}} - R_{\text{DINO}_{\text{ViT-B/16}}} - R_{\text{MoCo V3}_{\text{ViT-B/16}}} - R_{\text{MAE}_{\text{ViT-B/16}}}$ as in Fig. 3 (middle subfigure); (3) $R_{\text{DINO}_{\text{ViT-B/8}}} - R_{\text{DINO}_{\text{ViT-B/16}}} - R_{\text{MoCo V3}_{\text{ViT-B/16}}} - R_{\text{SwAV}_{\text{ResNet50}}} - R_{\text{MAE}_{\text{ViT-B/16}}}$ as in Fig. 3 (bottom subfigure). The full lists of classes are provided in the Appendix.

## 3.2 UNIFORM SAMPLING

Uniform sampling, a.k.a. random sampling, is an ordinary baseline to generate the subsets. Previous literature has been following this scheme to generate the subset with a uniform probability density function, e.g., randomly sample $N$ classes from the 1000 classes on ImageNet-1K. In Fig. 4, we exhibit the results of `RS-ID` from different self-supervised methods,

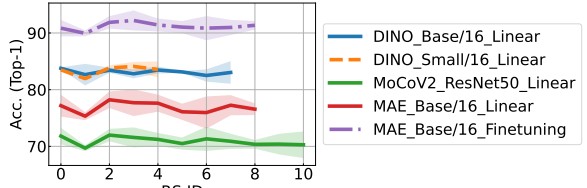

Figure 4: Illustration of accuracy comparison on different `RS` subsets. Each model is trained by three trials on both pre-training and linear probing/finetuning stages.

and derive the following discoveries: (i) It can be observed that the discriminability of weak and strong backbones is absent on DINO, as illustrated in Fig. 4 of black curves. (ii) Some trends are inconsistent, for instance, from subset `RS-3` to `RS-4`, the accuracy increases on DINO ViT-Base/16 while decreases on MAE and MoCo. These unreliabilities are as expected and inevitable since these subsets are randomly selected without any guidance or restriction.

## 3.3 PRIOR KNOWLEDGE

There are multiple prior knowledge that can be utilized as the policies to find out the winning tickets:

**Performance-Driven:** The performance of each class has a significant influence on reflecting the difficulties of classification, also showing the similarity measure on the latent feature space, so that it is crucial for leveraging the performance criterion of selecting the potential tickets. As shown in Fig. 5 (1), we use the pre-trained EfficientNet-L2 (Tan & Le, 2019) to sort the accuracy of per-class and select with four patterns: `PD-Top (easiest)`, `PD-Mid`, `PD-Bottom (hardest)`, and `PD-Uniform`. Some selected categories of images are shown in the ellipses of Fig. 5.

**WordNet Hierarchy:** In WordNet, the concepts are linked together in a hierarchy. This makes it easy to navigate between concepts. For instance, given a concept like motorcar, we can find out the concepts that are more specific, i.e., the immediate hyponyms. As shown in Fig. 5 (2), since the

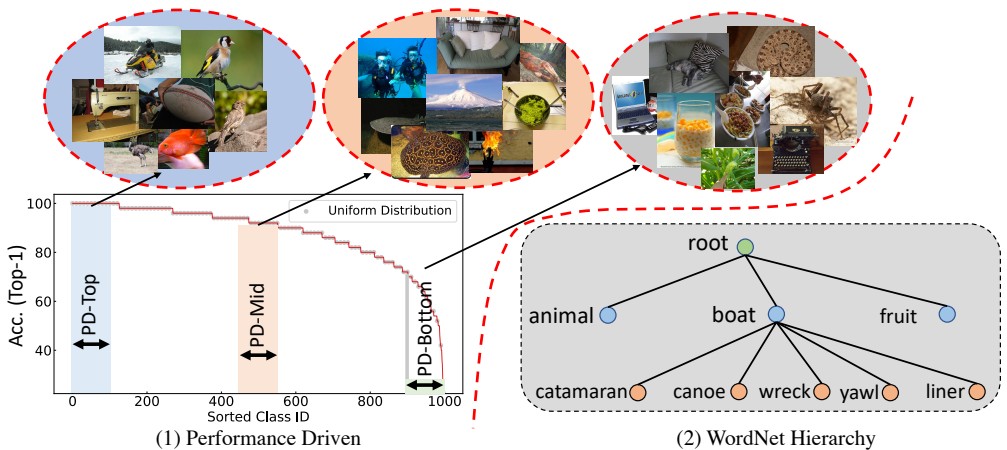

Figure 5: Illustration of the policies for *Performance Driven* (left) and *WordNet Hierarchy* (right). We also visualize images in the selected classes to reflect the difficulty level of the selected subsets.

classes of ImageNet-1K are the leaf nodes in the WordNet tree, we can merge leaf nodes that have a common parent and repeat this operation until obtaining enough coarse classes as the winning ticket.

**Word-embedding Clustering:** In this strategy, we aim to obtain the word-level embeddings of ImageNet-1K human-readable labels[3] and perform an unsupervised clustering method on them. Firstly, a pre-trained CLIP model (Radford et al., 2021) with ViT-L/14 (Dosovitskiy et al., 2020) backbone[4] with the property of context-dependency is employed to represent semantic labels through word embedding vectors. Next, both k-medoids an kmeans clustering algorithms are tested to group semantically similar labels in order to reduce the overlapping. Finally, different quality label features are utilized as the candidates in different clusters, and then the high-ranked semantic labels are picked out from all clusters to form the winning ticket.

**Semantic-embedding Clustering:** Instead of using the word-level embeddings of ImageNet-1K human-readable labels, we use the embedding by taking the average feature from a pre-trained DINO (Caron et al., 2021) model of ViT-B/8 for all images of that class in the validation set. Other procedures are the same as the above word-embedding clustering. This strategy is also applicable if the label is not available in the dataset.

| Policies | Subset Pool Abbreviation | | | |
|---|---|---|---|---|
| Random Sampling | $\{$RS-ID$\}$ (ID$\in \{0, 1, 2, \ldots, k_{\text{RS}}\}$) | | | |
| Empirical Risk Consistency | $\{$ERC-ID$\}$ (ID$\in \{0, 1, 2, \ldots, k_{\text{ERC}}\}$) | | | |
| Performance Driven | PD-Top | PD-Mid | PD-Bottom | PD-Uniform |
| WordNet Hierarchy | $\{$WNH-ID$\}$ (ID$\in \{0, 1, 2, \ldots, k_{\text{WNH}}\}$) | | | |
| Word Embedding Clustering | WEC | | | |
| Semantic Embedding Clustering | SEC | | | |
| Number Image Reduction | $\{$NiR-ID$\}$ (ID$\in \{0, 1, 2, \ldots, k_{\text{NiR}}\}$) | | | |

Table 1: Left column is different policies for identifying the desired dataset winning ticket. Right column is the abbreviations of the corresponding policies in the main text.

## 4 EXAMINING WINNING TICKETS ON VARIOUS ARCHITECTURES

We employ both Pearson Correlation Coefficient (PCC) (Pearson, 1895) ($\rho_p$) and Spearman's Rank Correlation (Spearman, 1961) ($\rho_s$) to examine the correlation of the performance between the full set and the selected winning ticket. The former metric focuses on the linear correlation of two input sets of variables, the latter one reflects the monotonic correlation of them. Here, we study the dataset lottery ticket hypothesis on different architectures and strategies used in practice. Specifically, we

---

[3]https://github.com/anishathalye/imagenet-simple-labels/blob/master/imagenet-simple-labels.json.

[4]BERT (Kenton & Toutanova, 2019) model is also tested in our ablation experiments.

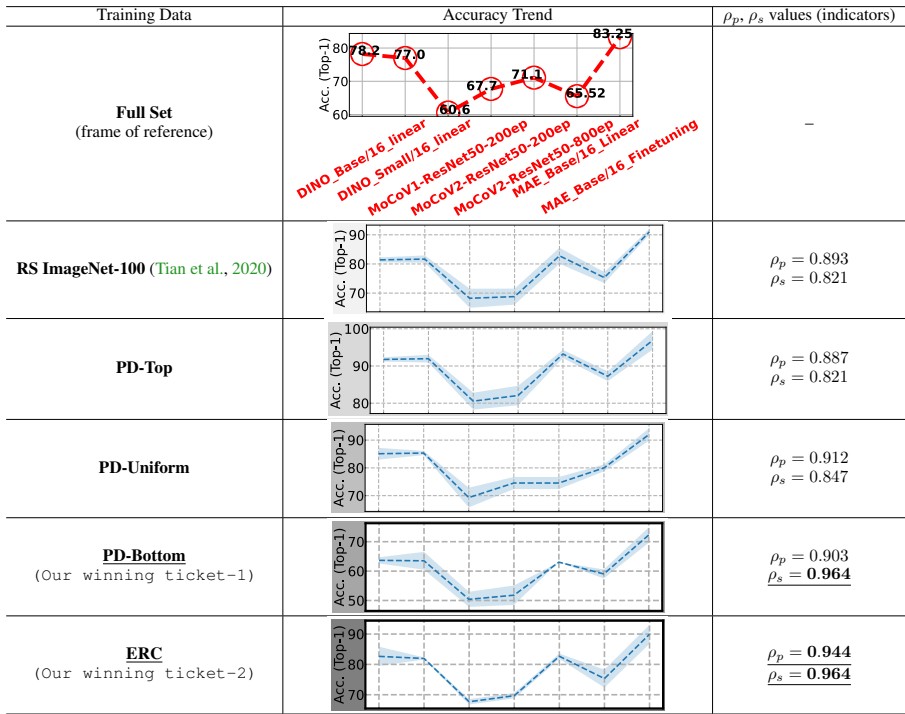

| Training Data | Accuracy Trend | $\rho_p, \rho_s$ values (indicators) |
|---|---|---|
| **Full Set** (frame of reference) | | – |
| **RS ImageNet-100** (Tian et al., 2020) | | $\rho_p = 0.893$ $\rho_s = 0.821$ |
| **PD-Top** | | $\rho_p = 0.887$ $\rho_s = 0.821$ |
| **PD-Uniform** | | $\rho_p = 0.912$ $\rho_s = 0.847$ |
| **PD-Bottom** (Our winning ticket-1) | | $\rho_p = 0.903$ $\rho_s = \mathbf{0.964}$ |
| **ERC** (Our winning ticket-2) | | $\rho_p = \mathbf{0.944}$ $\rho_s = \mathbf{0.964}$ |

Table 2: Overview of accuracy trends and correlation values (higher is better) across different frameworks. The first row represents the accuracy on full data, in other rows, each is corresponding to one subset. As limited by the pages, more subsets' results will be provided in the Appendix.

consider the typical deep neural architectures of residual networks (He et al., 2016) and vision transformers (Dosovitskiy et al., 2020).

**ResNet.** We examine the dataset lottery ticket hypothesis as applied to the typical ConvNet of ResNet-50 (He et al., 2016). We use MoCo V1, V2 as the training methods with 200 and 800 epochs. All of our training settings follow their default design, as well as the following vision transformer models.

**Vision Transformer.** We assess the dataset lottery ticket hypothesis on ViT models. We use the ViT-Base and ViT-Small architectures with MAE and DINO frameworks. The MAE model is trained with 800 epochs and DINO ViT-Base, ViT-Small are trained with 400, 800 epochs, respectively.

**Results.** As shown in Table 2, `PD-Top` indicates that we choose the best-performed classes according to a pre-trained model, i.e., the easiest categories. Our results show that this subset has the worst consistency on linear correlation. We can also see the first two subsets have the same $\rho_s$, which means that they enjoy the same property of ranked performance. Moreover, choosing the hardest classes (`PD-Bottom`) has good correlations to the full data and is one of our winning tickets. The correlation among the 7 framework configurations in Table 2 is the most crucial indicator for evaluation to reflect the quality of the selected subset, as our goal of this paper is to identify the proper subset that has the same correlation or performance trend as on the full data. Thus, for some heavy and costly tasks like ablation study, architecture search, etc., it is more practical and efficient for us to employ the selected subset instead of the full data to explore the best configuration that we desired. The results of `WEC`, `SEC` and `WNH` are similar to `PD-Uniform` so they are omitted in Table 2 and given in Appendix.

## 4.1 CONSISTENCY

**Consistency on Data Augmentation.** To assess the consistency of performance on different data augmentations, we use MoCo V1 and V2 which only contain the difference on data augmentation. The results are shown in Fig. 6 (1), it is surprising to see that nearly *all* the subsets are not sensitive

to the data augmentation factor, obtaining similar accuracy on MoCo V1 and V2, but are generally still consistent on performance.

**Consistency on Training Budgets.** On the self-supervised task, it is known that more training budget can substantially achieve better performance. In this subsection, we examine whether this property still exists on the dataset winning ticket. The results are shown in Fig. 6 (2).

**Consistency on Evaluation Strategies.** To assess the consistency of performance on different evaluation strategies, we employ MAE with finetuning and linear probing evaluations. The results are shown in Fig. 6 (3), it can be observed that the performance on all subsets is consistent.

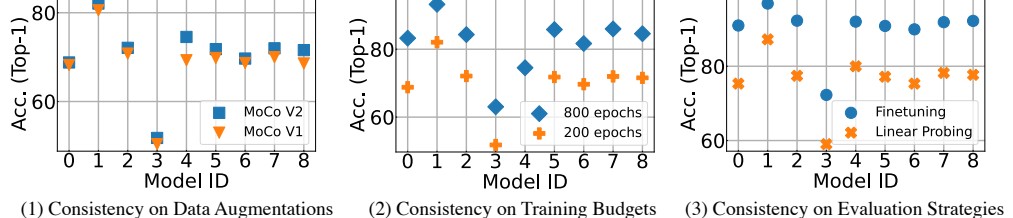

(1) Consistency on Data Augmentations     (2) Consistency on Training Budgets     (3) Consistency on Evaluation Strategies

Figure 6: Illustration of the consistency conditions for *Data Augmentation* (left), *Training Budgets* (middle) and *Evaluation Strategies* (right). In each subfigure, the abscissa axis represents the used subsets: CMC subset; PD-Top; PD-Mid; PD-Bottom; PD-Uniform; RS-1; RS-2; RS-3; RS-4.

## 4.2 GENERALIZABILITY

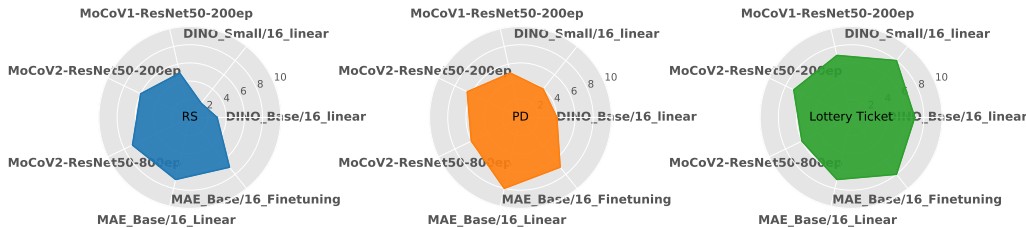

Figure 7: Illustration of the polygon radar chart for random sampling, PD-Uniform and our lottery ticket on several indicator approaches. Our lottery ticket has the best ability across these indicators.

It is interesting to see the generalizability of the identified winning tickets on those training methods outside the selection policies. Our conducted experiments above actually involved this verification: The per-class empirical risk consistency does not employ MoCo V1 and V2 but from our results in Table 2 it seems the winning tickets can still handle them well. Since MoCo V2 performs heavier data augmentation and maintains the other training settings the same as V1, this result demonstrates that the winning ticket is still a good indicator and can reflect the performance trend on input dimension when the transformations are changed. Besides the input transformations, i.e., MoCo V1 to V2, we also include two more factors: **(i)** batch size effect, and (ii) stochastic regularization enabled by DropPath (Huang et al., 2016). We study how these subtle changes could affect the generalizability on winning ticket of the self-supervised models, the results and discussions are in our Appendix. To assess the generalizability beyond the visual datasets, we further provide the preliminary results on the AG's News Topic Classification Dataset (Zhang et al., 2015) with the comparison between the random subset and lottery ticker subset in Appendix I.

## 5 DISCUSSIONS AND FUTURE WORK

We have introduced dataset lottery ticket hypothesis (**DLTH**), an initial study to investigate whether there exists winning tickets (subsets) that can reflect the consistency on performance when performing different self-supervised approaches and hyperparameters. Through extensive empirical evaluations, we conclude that the randomly generated subset is not qualified for indicating the training settings and configurations of full data. We proposed several policies by incorporating prior knowledge to find out the desired winning tickets. In our experiments, the following trends are consistent: **(i)** Random subsets are not guaranteed to obtain consistent results, in most cases they are unreliable. **(ii)** Prior knowledge (e.g., the difficulties of classes) can substantially improve the quality of the selected winning tickets which makes the selected ticket more aligned to the full data.

**(iii)** Two drawbacks of subsets are noticed, even on the winning tickets: (1) subsets are not sensitive to different backbones if the training frameworks are strong, such as DINO; (2) subsets are not much sensitive to data augmentations, such as the case from MoCo V1 to V2.

**Limitations and Future Work.** Currently, we only verified our hypothesis on the vision domain of ImageNet-1K modality, we do not investigate it on more modalities such as video, text, etc. Considering that different disciplines may have different characteristics or properties, for the future work, we hope our hypothesis can be proven in other domains such as NLP and speech datasets.

## POTENTIAL ETHICAL IMPACT

As machine learning algorithm has been well studied in data-intensive applications, such as the large-scale ImageNet-1K classification, but is often hampered when the training data is small, this work leveraging subset of data may have the following potential positive and negative impacts in the society. For the positive impacts, the proposed dataset lottery ticket scheme can significantly help save training time, computational resources and energy for tuning hyperparameters of heavy models, which is highly eco-friendly. However, for the negative impacts, as this work tackles this problem through involving prior knowledge for identifying dataset winning tickets, the model may be biased on the training subset if the full data is unbalanced. Thus, we should be cautious on the result of failure from the system which could cause unreliable conclusions, such as the unbalanced dataset like medical images and further was misleading in some particular domains.

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

APPENDIX

In the appendix, we provide details omitted in the main text, including:

• Section A: Exploration experiments of more classes or more samples in per-class. (Section 4 "Examining Winning Tickets on Various Architectures" of the main paper.)

• Section B: Results of dataset lottery ticket hypothesis on supervised learning. (Section 4 "Examining Winning Tickets on Various Architectures" of the main paper.)

• Section C: An introduction of experimental settings. (Section 4 "Examining Winning Tickets on Various Architectures" of the main paper.)

• Section D: Full lists of winning tickets on ImageNet-1K. (Section 3 "Winning Tickets Policies" of the main paper.)

• Section E: More results of different subsets. (Table 2 "Overview of accuracy trends and correlation values." of the main paper.)

• Section F: More results and discussions on the study of generalizability. (Section 4.2 "Generalizability" of the main paper.)

• Section G: Background of our work. (Section 3 "Winning Tickets Policies" of the main paper.)

• Section H: Implementation details and experimental settings. (Section 4 "Examining Winning Tickets on Various Architectures" of the main paper.)

• Section I: Results on AG's news topic classification dataset. (Section 4.2 "Generalizability" of the main paper.)

## A  MORE CLASSES OR MORE SAMPLES IN PER-CLASS?

Instead of selecting a subset along with the class, we can also keep all categories and reduce the number of images in each class (`NiR`). This strategy will make the subset more diverse but the same category has fewer training samples. In this section, we examine the influence of number of images in each class for the self-supervised learning. To study this, we keep the set of

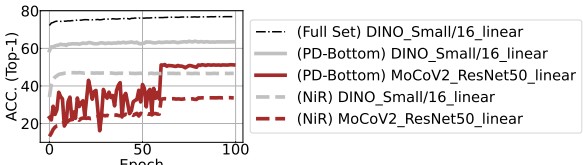

Figure 8: Comparison of linear probing accuracy on `Full Set`, `PD-Bottom` and `NiR` subsets.

original classes and randomly select 1/10 of the images in each class. We illustrate the loss evolution of self-supervised pre-training and linear probing in Fig. 8. It can be observed that the strategy of `NiP` has low accuracy of 38% on MoCo V2 and 46% on DINO (we use `Full Set` and the *low absolute performance* subset `PD-Bottom` as comparisons.). Though it is not necessary the performance is low since we focus more on the trend across the frameworks, the model is concerned to be "dull" and may be biased to some particular classes that contain few samples since samples in each class are insufficient.

## B  DATASET LOTTERY TICKET HYPOTHESIS ON SUPERVISED LEARNING

We examine the dataset lottery ticket hypothesis on *supervised learning* with the following networks: ResNet-50, ResNeXt50 (32×4d), RegNet_Y_3_2GF, DenseNet121, MobileNet_V3_Large, ConvNext_Small on the subsets of `NiR`, `PD-Top`, `PD-Mid`, `PD-Bottom`, `PD-Uniform`, `ERC` and `RS`. We did not use ViT here as it is hungry for data in supervised learning. The results are shown in Table 3, which are aligned with those on self-supervised learning task. The slight difference we observe is that on supervised learning, `PD-Uniform` performs much better than it on self-supervised learning. The common conclusion is that the subset from the dimension of sample's number `NiR` and random sampling subset `RS` are mediocre with a weak alignment to the full data.

| Training Data | Accuracy Trend (Top-1) | | | | | | $\rho_p, \rho_s$ values |
|---|---|---|---|---|---|---|---|
| | ResNet-50 | ResNeXt50_32X4D | RegNet_Y_3_2GF | DenseNet121 | MobileNet_V3_L | ConvNext_S | |
| NiR | 51.12 | 52.34 | 53.42 | 52.45 | 45.71 | 29.47 | $\rho_p = 0.637, \rho_s = 0.886$ |
| RS | 81.04 | 85.36 | 85.52 | 84.84 | 79.94 | 81.27 | $\rho_p = 0.922, \rho_s = 0.886$ |
| PD-Top | 90.36 | 91.75 | 93.83 | 93.33 | 89.87 | 91.03 | $\rho_p = 0.926, \rho_s = 0.943$ |
| PD-Mid | 83.53 | 83.12 | 86.97 | 86.14 | 83.73 | 83.11 | $\rho_p = 0.885, \rho_s = 0.943$ |
| PD-Bottom | 62.29 | 61.84 | 64.71 | 65.01 | 60.21 | 60.00 | $\rho_p = 0.918, \rho_s = 0.829$ |
| PD-Uniform | 83.27 | 84.51 | 87.49 | 86.53 | 82.05 | 83.19 | $\rho_p = 0.959, \rho_s = 0.943$ |
| ERC-100 | 84.79 | 85.81 | 89.31 | 88.32 | 83.75 | 84.65 | $\rho_p = 0.945, \rho_s = 1.000$ |
| ERC-155 | 82.37 | 85.19 | 87.53 | 86.21 | 80.75 | 82.49 | $\rho_p = 0.971, \rho_s = 0.943$ |

Table 3: Accuracy trends and correlation values (higher is better) on supervised learning across different backbone networks.

## C MORE DETAILS OF SUBSET GENERATION EXPERIMENTS

Considering that if the size of the subset is too large, the subset will lose its role and advantage of fast hyperparameter searching. To this end, in the policy of per-class empirical consistency, $T$ is set to 200 and $M_{\text{target}}$ is set to 100. It means that if the size of selected subset is large than 200, we will randomly choose 100 classes among them as the new subset, so that we can guarantee the final scales of subsets will not be larger than $T$. Otherwise, the subset will keep the originally identified classes.

## D FULL NAMES OF SUBSET LISTS ON IMAGENET-1K

### D.1 PD-BOTTOM (WINNING TICKET 1)

```
n01667778 n01693334 n01729977 n01740131 n01744401 n01753488
n01756291 n01773549 n01773797 n01775062 n02088466 n02089973
n02096294 n02106030 n02107908 n02109961 n02110185 n02114712
n02119022 n02123045 n02123159 n02124075 n02167151 n02229544
n02395406 n02403003 n02412080 n02415577 n02441942 n02443114
n02443484 n02493793 n02497673 n02669723 n02749479 n02769748
n02776631 n02808440 n02895154 n02974003 n02979186 n02988304
n02999410 n03016953 n03045698 n03125729 n03146219 n03179701
n03249569 n03461385 n03485407 n03642806 n03657121 n03658185
n03710637 n03710721 n03770679 n03773504 n03782006 n03787032
n03793489 n03832673 n03866082 n03871628 n03895866 n03950228
n03976657 n04008634 n04081281 n04152593 n04239074 n04264628
n04285008 n04286575 n04355933 n04356056 n04357314 n04380533
n04392985 n04428191 n04443257 n04493381 n04525038 n04557648
n04560804 n04589890 n04591157 n04592741 n04599235 n07579787
n07584110 n07734744 n07860988 n07892512 n07930864 n09332890
n09399592 n09428293 n12144580 n13133613
```

### D.2 ERC (WINNING TICKET 2)

```
n01484850 n01498041 n01580077 n01601694 n01608432 n01622779
n01669191 n01685808 n01688243 n01694178 n01698640 n01728920
n01755581 n01770393 n01824575 n01829413 n01855032 n01873310
n01914609 n01978287 n01981276 n01983481 n01985128 n01990800
n02002556 n02007558 n02012849 n02056570 n02074367 n02077923
n02088238 n02090721 n02091032 n02091467 n02093859 n02096437
n02097209 n02098105 n02099849 n02100583 n02101006 n02108089
n02110806 n02111277 n02111500 n02111889 n02114548 n02128385
n02177972 n02190166 n02346627 n02356798 n02363005 n02444819
n02445715 n02481823 n02484975 n02486261 n02641379 n02690373
n02786058 n02797295 n02799071 n02804414 n02807133 n02823428
n02843684 n02865351 n02892201 n02948072 n02977058 n02992211
n03000684 n03018349 n03062245 n03110669 n03131574 n03207941
n03388183 n03394916 n03445924 n03447721 n03450230 n03459775
n03482405 n03538406 n03584829 n03595614 n03598930 n03649909
```

```
n03697007  n03764736  n03786901  n03873416  n03876231  n03877472
n03877845  n03888605  n03891251  n03903868  n03929855  n03935335
n03938244  n03967562  n03977966  n03998194  n04005630  n04033901
n04039381  n04041544  n04099969  n04153751  n04209133  n04238763
n04251144  n04273569  n04277352  n04311174  n04336792  n04355338
n04367480  n04371774  n04417672  n04418357  n04442312  n04465501
n04467665  n04482393  n04485082  n04487394  n04509417  n04515003
n04517823  n04523525  n04532106  n04542943  n04553703  n04554684
n04579432  n04589890  n04597913  n04604644  n07565083  n07615774
n07693725  n07697537  n07717556  n07718472  n07753275  n07873807
n07875152  n09229709  n12620546  n12768682  n13052670
```

## E    MORE RESULTS

More results of accuracy trends and correlation values are shown in Fig. 4.

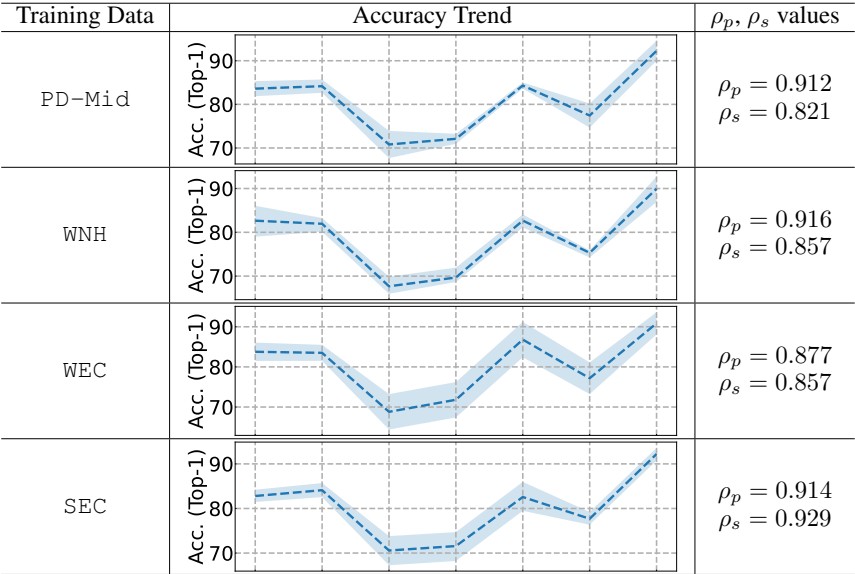

| Training Data | Accuracy Trend | $\rho_p, \rho_s$ values |
|---|---|---|
| PD-Mid | | $\rho_p = 0.912$ $\rho_s = 0.821$ |
| WNH | | $\rho_p = 0.916$ $\rho_s = 0.857$ |
| WEC | | $\rho_p = 0.877$ $\rho_s = 0.857$ |
| SEC | | $\rho_p = 0.914$ $\rho_s = 0.929$ |

Table 4: More accuracy trends and correlation values (higher is better) across different frameworks.

## F    MORE ANALYSES ON GENERALIZABILITY

**Settings: (i)** Batch size effect. On this factor, we employ *MAE_Base/16_Linear* as the baseline and examine the consistency between the winning ticket and full data. The default batch size in MAE framework is 4,096, we test 2,048 and 1,024 respectively for studying the generalizability of our selected winning ticket. **(ii)** Stochastic regularization with DropPath. On this factor, we employ *DINO_Small/16_linear* as the baseline, the default DropPath ratio in DINO framework is 0.1, we test additional 0.3, 0.5, respectively. Our results indicate that the selected subsets have good generalizability on slightly different training settings.

### F.1    A BRIEF THEORY ANALYSIS OF DATASET LOTTERY TICKET HYPOTHESIS

Inspired by (Sorscher et al., 2022), we also use a teacher-student perceptron scheme to explain our DLTH theoretically. We consider select the subset training dataset (the subset has $P$ examples with $\{\mathbf{x}^\mu, y^\mu\}_{\mu=1,...,P}$, where $\mathbf{x}^\mu \sim \mathcal{N}(0, I_N)$) by keeping only the examples with the smallest margin $|z^\mu| = |\mathbf{S}_{\text{probe}} \cdot \mathbf{x}^\mu|$ along a probe student $\mathbf{S}_{\text{probe}}$, which matches one of our proposed selection policies. The selected subset will follow the distribution $p(z)$ along the direction of $\mathbf{S}_{\text{probe}}$. Similarly, we can obtain the generalized theory for an arbitrary data distribution $p(z)$ for the small-margin

selection strategy. $\mathbf{S}_{\text{probe}}$ contains overlap with the teacher that is determined by the angle $\theta = \cos^{-1}\left(\frac{\mathbf{S}_{\text{probe}} \cdot \mathbf{T}}{\|\mathbf{S}_{\text{probe}}\|_2 \|\mathbf{T}\|_2}\right)$.

After the subset has been selected, we train a new student $S$ from scratch on the selected subset. The typical learning algorithm is to train a classifier for the training data with the maximal margin $\kappa = \min_\mu \mathbf{S} \cdot (y^\mu \mathbf{x}^\mu)$ to find the optimized $S$. Different from data pruning of computing the generalization error $\varepsilon_g$ of this student, which is governed by the overlap between the student and teacher, $\varepsilon_g = \cos^{-1}(R)/\pi$, where $R = \mathbf{S} \cdot \mathbf{T}/\|\mathbf{S}\|_2 \|\mathbf{T}\|_2$. we focus on the consistency of test error $\varepsilon s$ across different configurations for the final perceptron as a function.

# G  BACKGROUND

**Self-supervised Learning.** Self-supervised pre-training has been recognized as a promising technique in various fields, such as Computer Vision (Caron et al., 2021; 2020; He et al., 2022), NLP (Kenton & Toutanova, 2019; Yang et al., 2019; Liu et al., 2019; Brown et al., 2020), Speech (Hsu et al., 2021; Liu et al., 2021; Chen et al., 2022) and Multimodality (Radford et al., 2021; Ramesh et al., 2021; 2022). The major drawback of this technique is the costly training overhead, which dramatically impedes more investigation and exploration. For instance, GPT-3 model (Brown et al., 2020) contains 175 billion parameters and is trained roughly requiring 1,024 A100 GPUs for more than one month, the estimated training cost of it is at least \$4.6 million. In the vision domain, DINO ViT-B/8 (Caron et al., 2021) is also trained on 176 GPUs which is inaccessible for most research institutions.

**Efficient Hyperparameter Tuning.** The most common hyperparameter tuning method is the *grid search*. While it is always in low-efficiency when the number of hyperparameters is large, as well as the big training data and model size. *Random search* with human experience and *Bayesian optimization* (Bergstra et al., 2011; Snoek et al., 2012; Hutter et al., 2019) are two popular fast hyperparameter tuning methods and have been successfully applied to many applications in practice. In this study, we focus on finding a proper subset (winning ticket) from the full data, which is orthogonal to these black-box strategies and can be utilized with them simultaneously. Moreover, a suitable subset is more efficient if the model's training cost is high.

**Model-level Lottery Ticket Hypothesis.** The original lottery ticket hypothesis (Frankle & Carbin, 2019) and its follow-ups (Zhou et al., 2019; Evci et al., 2020; Wang et al., 2020; Chen et al., 2020a; Morcos et al., 2019; You et al., 2019; Savarese et al., 2020) are proposed on the neural network dimension as the model pruning technique. The core of this hypothesis is that a randomly-initialized neural network contains a subnetwork that can match the test accuracy of the original model after training for the same number of iterations. PrAC (Zhang et al., 2021) observed that a sparse model can be explored with training and pruning the dense network on the compact PrAC set, which has a different goal from the proposed DLTH. In this work, we study the lottery ticket hypothesis on the *datatset* dimension to reflect the consistency of performance trend, and propose several practice ways to identify the dataset winning tickets.

**Active Learning and Dataset Pruning.** Active learning (Settles, 2009) is similar to the semi-supervised learning strategy by involving the human participation, and the process of selecting suitable candidate sets for manual labeling through machine learning algorithms. The idea of active learning is to obtain the sample that is more difficult to classify through ML approach, then, let human reconfirm and review them. After that, we can re-use the manually labeled data with a supervised learning or semi-supervised learning model. There are many recent literature following this interesting paradigm (Ash et al., 2019; Mirzasoleiman et al., 2020; Citovsky et al., 2021). In contrast, DLTH focuses on selecting a subset with consistent performance trends as the full data. For the dataset pruning, Sorscher et al. (Sorscher et al., 2022) focused on the scaling of testing error with dataset size and proposed a self-supervised pruning metric to demonstrate comparable results to the supervised metrics. Different from the dataset pruning setting, in our scenario, we do not care about the absolute accuracy of the individual model trained on the subset but the consistency to the models trained on full data. The two tasks essentially have different goals, and our pruning ratio on the original dataset is also much higher than dataset pruning.

# H    IMPLEMENTATION DETAILS AND EXPERIMENTAL SETTINGS

## H.1    DINO (CARON ET AL., 2021) PRETRAINING AND LINEAR PROBING

We conduct experiments two configurations of DINO framework (Caron et al., 2021): (I) `DINO_Base/16_linear` and (II) `DINO_Small/16_linear`. The **pre-training** settings are provided in Table 5. The **linear probing** settings are in Table 6.

| arch. | vit_small | vit_base |
|---|---|---|
| optimizer | AdamW | AdamW |
| patch_size | 16 | 16 |
| out_dim | 65,536 | 65,536 |
| norm_last_layer | false | true |
| warmup_teacher_temp | 0.04 | 0.04 |
| teacher_temp | 0.07 | 0.07 |
| warmup_teacher_temp_epochs | 30 | 50 |
| momentum_teacher | 0.996 | 0.996 |
| use_bn_in_head | false | false |
| drop_path_rate | 0.1 | 0.1 |
| use_fp16 | false | false |
| weight_decay | 0.04 | 0.04 |
| weight_decay_end | 0.4 | 0.4 |
| clip_grad | 0.0 | 0.3 |
| batch_size | 1024 | 1024 |
| epochs | 800 | 400 |
| freeze_last_layer | 1 | 3 |
| lr | 0.0005 | 0.00075 |
| warmup_epochs | 10 | 10 |
| min_lr | 2e-05 | 2e-06 |
| global_crops_scale | [0.25, 1.0] | [0.25, 1.0] |
| local_crops_scale | [0.05, 0.25] | [0.05, 0.25] |
| local_crops_number | 10 | 10 |
| seed | 0 | 0 |

Table 5: **DINO (Caron et al., 2021) pre-training setting on subsets.**

| arch. | vit_small | vit_base |
|---|---|---|
| patch_size | 16 | 16 |
| n_last_blocks | 4 | 1 |
| avgpool_patchtokens | false | true |
| checkpoint_key | teacher | teacher |
| epochs | 100 | 100 |
| lr | 0.001 | 0.001 |
| batch_size | 1,024 | 1,024 |
| optimizer | SGD | SGD |
| weight_decay | 0.0 | 0.0 |
| optimizer_momentum | 0.9 | 0.9 |
| learning_rate_schedule | cosine decay | cosine decay |

Table 6: **DINO (Caron et al., 2021) linear probing setting on subsets.**

## H.2    MOCO (HE ET AL., 2020) PRETRAINING AND LINEAR PROBING

We conduct experiments with three configrations on MoCo V1&V2 (He et al., 2020; Chen et al., 2020b) frameworks: (I) `MoCoV1-ResNet50-200ep`, (II) `MoCoV2-ResNet50-200ep` and (III) `MoCoV2-ResNet50-800ep`. The details of **pre-training** and **linear probing** settings are provided in Table 7 and Table 8, respectively.

| config. | C1 | C2 | C3 |
|---|---|---|---|
| arch. | ResNet50 | ResNet50 | ResNet50 |
| batch_size | 256 | 256 | 256 |
| epochs | 200 | 200 | 800 |
| lr | 0.03 | 0.03 | 0.03 |
| learning_rate_schedule | cosine decay | cosine decay | cosine decay |
| optimizer | SGD | SGD | SGD |
| momentum | 0.9 | 0.9 | 0.9 |
| weight_decay | 1e-4 | 1e-4 | 1e-4 |
| moco-dim | 128 | 128 | 128 |
| moco-k | 65,536 | 65,536 | 65,536 |
| moco-m | 0.999 | 0.999 | 0.999 |
| moco-t | 0.07 | 0.2 | 0.2 |
| mlp_head | false | true | true |
| aug_plus | false | true | true |

Table 7: **MoCo (He et al., 2020) pre-training setting on subsets.**

| config. | C1 | C2 | C3 |
|---|---|---|---|
| arch. | ResNet50 | ResNet50 | ResNet50 |
| batch_size | 256 | 256 | 256 |
| epochs | 100 | 100 | 100 |
| lr | 30 | 30 | 30 |
| learning_rate_schedule | [60, 80] | [60, 80] | [60, 80] |
| optimizer | SGD | SGD | SGD |
| momentum | 0.9 | 0.9 | 0.9 |
| weight_decay | 0 | 0 | 0 |
| augmentation | RandomResizedCrop | RandomResizedCrop | RandomResizedCrop |

Table 8: **MoCo (He et al., 2020) linear probing setting on subsets.**

### H.3   MAE (HE ET AL., 2022) PRETRAINING, FINETUNING AND LINEAR PROBING

We conduct experiments on MAE (He et al., 2022) with one pretraining configuration and two evaluation protocols: (I) MAE_Base/16_Linear and (II) MAE_Base/16_Finetuning.

**Pre-training:** The setting used for our experiments is in Table 9. Following the default protocal, we do *not* use color jittering, drop path, or gradient clip. We also use the linear *lr* scaling rule: *lr = base_lr*×batchsize / 256. **End-to-end fine-tuning:** The setting used in our experiments is provided in Table 10. **Linear probing:** Our linear classifier training setting is provided in Table 11.

| arch. | vit_base |
|---|---|
| optimizer | AdamW |
| base_learning_rate | 1.5e-4 |
| weight_decay | 0.05 |
| optimizer_momentum | $\beta_1, \beta_2 = 0.9, 0.95$ |
| batch_size | 4,096 |
| learning_rate_schedule | cosine decay |
| warmup_epochs | 40 |
| augmentation | RandomResizedCrop |

Table 9: **MAE (He et al., 2022) pre-training setting on subsets.**

## I   RESULTS ON AG'S NEWS TOPIC CLASSIFICATION DATASET

**Training and Evaluation Settings.** To further examine the generalizability of the proposed DLTH, we extend our method to the NLP domain to see whether this hypothesis still holds using AG's news topic classification dataset (Zhang et al., 2015). This dataset is widely used as a text classification benchmark. It consists of 4 coarse classes from the original corpus. Each class contains 30K training samples and 1.9K testing samples. The total number of training samples is 120K and testing 7.6K.

| arch. | vit_base |
|---|---|
| optimizer | AdamW |
| base_learning_rate | 1e-3 |
| weight_decay | 0.05 |
| optimizer_momentum | $\beta_1, \beta_2=0.9, 0.999$ |
| layer-wise_lr_decay | 0.75 |
| batch_size | 1,024 |
| learning_rate_schedule | cosine decay |
| warmup_epochs | 5 |
| training_epochs | 100 |
| augmentation | RandAug (9, 0.5) |
| label_smoothing | 0.1 |
| mixup | 0.8 |
| cutmix | 1.0 |
| drop_path | 0.1 |

Table 10: **MAE (He et al., 2022) end-to-end fine-tuning setting on subset.**

| arch. | vit_base |
|---|---|
| optimizer | LARS |
| base_learning_rate | 0.1 |
| weight_decay | 0 |
| optimizer_momentum | 0.9 |
| batch_size | 16,384 |
| learning_rate_schedule | cosine decay |
| warmup_epochs | 10 |
| training_epochs | 90 |
| augmentation | RandomResizedCrop |

Table 11: **MAE (He et al., 2022) linear probing setting on subsets.**

**Configurations and Results.** We explore four configurations as shown in Table 12. The results are provided in Table 13. For the lottery ticket subset, we simply select consistent samples across four configurations until reaching 10% of the full data. We do this twice to construct different lottery ticket subsets to mitigate variance in different runs. We can observe that the models trained on lottery ticket subsets are much more robust with higher correlation values to the full data than the modles trained on the random subset baselines.

| config. | C1 | C2 | C3 | C4 |
|---|---|---|---|---|
| batch_size | 64 | 64 | 128 | 64 |
| epochs | 10 | 10 | 10 | 10 |
| lr | 5 | 5 | 5 | 10 |
| learning_rate_schedule | StepLR | CosineAnnealingLR | StepLR | StepLR |
| optimizer | SGD | SGD | SGD | SGD |

Table 12: Training configurations on AG's news topic classification dataset.

| Training Data | Accuracy Trend | | | | $\rho_p$ value |
|---|---|---|---|---|---|
| | C1 | C2 | C3 | C4 | |
| Full | 0.910 | 0.905 | 0.896 | 0.909 | – |
| RS-0 | 0.864 | 0.862 | 0.862 | 0.863 | 0.764 |
| RS-1 | 0.863 | 0.867 | 0.863 | 0.871 | 0.437 |
| RS-2 | 0.863 | 0.870 | 0.859 | 0.868 | 0.589 |
| RS-3 | 0.870 | 0.868 | 0.863 | 0.864 | 0.617 |
| RS-4 | 0.861 | 0.862 | 0.860 | 0.867 | 0.555 |
| Lottery Ticket-0 | 0.871 | 0.864 | 0.863 | 0.867 | **0.814** |
| Lottery Ticket-1 | 0.870 | 0.865 | 0.864 | 0.871 | **0.863** |

Table 13: Accuracy trends and correlation values on AG's news classification dataset.