# OpenReview forum: "Does Dataset Lottery Ticket Hypothesis Exist?"
_ICLR.cc/2023/Conference — Submitted to ICLR 2023_

### Official Review · Reviewer_gNgq · 2022-10-24

**Confidence:** 3
**Correctness:** 3
**Technical Novelty And Significance:** 2
**Empirical Novelty And Significance:** 2
**Recommendation:** 3

**Clarity, Quality, Novelty And Reproducibility:**

Clarity: The writing of this paper is overall easy to understand but the problem definitions and the experiment settings are confusing.
Novelty: The data lottery ticket problem is not so different from data pruning though following a different motivation. The methods to choose winning tickets seem novel to me.
Reproducibility: The reproducing details are missing.

**Strength And Weaknesses:**

Strengths:
1. Data lottery ticket is an important issue and investigating the lottery ticket problem from the view of training data is quite novel. Although data pruning often achieves the same effect as the data lottery ticket, data pruning has a different motivation that aims at removing biases and noises.

Weaknesses:
1. The definition of the data lottery ticket problem is ambiguous. The author stated in Definition 1 that "a subset that has the same or similar empirical behaviors and performance trends as the original full dataset when performing different training approaches and hyperparameters on it".
-  It lacks an elaboration on "performance trends" which can be pre-trained performance, training error, generalization error, downstream performance, etc.
-  The performance trends involve the selection of models. However, different pools of models may have different performance result trends. The author didn't give a clear definition as to how these training structures and training approaches should be chosen.
- A formal definition could be given instead of a text description for better clarity.

2. The motivation for choosing a lottery ticket with a consistency principle is ill-justified. For example, the authors stated in 3.2 that " Some trends are inconsistent, for instance, from subset RS-3 to RS-4, the accuracy increases on DINO ViT-Base/16 while decreases on MAE and MoCo" and use this to support their claim that uniform sampling lacks consistency. However, the models are tested via linear probing and the increase in model hyperparameters may result in overfitting or instability during fine-tuning and lead to these performance drops. Overall, this inconsistency may be the result of the difficult balancing of model size and subset size. It does not necessarily reflect the trend of the expressiveness of the models.

3. I'm doubtful about the training settings. The authors use full-dataset trained backbones to calculate test accuracies and only test if the performance trends are consistent. In most applications, users are more concerned about the performance achieved by directly tuning their model on the subsets. The paper lacks a discussion on this more practical setting. It seems that section 4.2 discusses this subject but only approximate numbers are provided in the radar char and lack precise results.

4. The class semantic-based methods for choosing lottery tickets are hard to apply beyond labeled image datasets. For example, many pre-trained models are using crawled images without labels. In these cases, the authors would not have the label semantics that is crucial for constructing lottery ticket subsets.

5. The experiment is overall imprecise and missing important settings.
- Most results are provided as visualization and charts, which are not precise enough. Eg., in Figure 4, the trend of DINO_base and DINO_small is plotted as line chart (performance vs set ID ), yet the scale for set ID is set to 2, making analyzing the trends between RS-3 and RS-4 difficult. Also, in Figure 7, no precise number is given and it is confusing what the scale "2,4,6,8,10" means.
- I'm not convinced by the results that inconsistency is an issue. As shown in Table 2, the Pearson correlation reaches 0.893 using random sampling subsets. After using some simple subset selection procedures, this consistency rises to 0.944.
- The training accuracy for direct learning on the subsets is missing, as stated in 3.
- The reproducing details are missing. I can't find the code or the hyperparameters for these experiments.



**Summary Of The Paper:**

This paper introduces the data lottery ticket hypothesis, which intends to select a subset that matches the performance of the original dataset. This hypothesis, if it works, can reduce training costs and democratize self-supervised pretraining. The authors first suggest that a good data lottery ticket subset should match the original performance in training, generalization, and model consistency. Based on these principles, the authors question the random sampling approach and propose the policies of Empirical Risk Trend and incorporate prior knowledge to generate the winning tickets. The analysis is demonstrated with a suite of self-supervised experiments with various structures (DINO, MAE, MOCOV).



**Summary Of The Review:**

I lean to reject this paper mainly because (1) the consistency principle is not well justified and could be much influenced by subset size and model structures, (2) the algorithm is heavily dependent on class labels and semantics, which may not be available in many datasets, including most NLP tasks where self-training is most heavily used (3) the overall presentation of the paper can be more organized and polished, (4) the reproducing details and the codes for the experiments are missing.

---

> ### Author Response · Authors · 2022-11-16
> **Response to Reviewer gNgq (3/3)**
>
> &nbsp;
> >The class semantic-based methods for choosing lottery tickets are hard to apply beyond labeled image datasets. For example, many pre-trained models are using crawled images without labels. In these cases, the authors would not have the label semantics that is crucial for constructing lottery ticket subsets.
>
> A: We clarify the *semantic-based methods* DO NOT use any human-annotated labels since we use **a pre-trained DINO model (DINO is a self-supervised learning approach) to extract representations and then cluster them in an unsupervised manner**. As we have stated in this part of the paper, this strategy is applicable if the label is not available in the dataset.
> &nbsp;
> >The experiment is overall imprecise and missing important settings.
>
> >Most results are provided as visualization and charts, which are not precise enough. Eg., in Figure 4, the trend of DINO_base and DINO_small is plotted as line chart (performance vs set ID ), yet the scale for set ID is set to 2, making analyzing the trends between RS-3 and RS-4 difficult. Also, in Figure 7, no precise number is given and it is confusing what the scale "2,4,6,8,10" means.
>
> A: Thanks for the suggestion. We list the numerical results of Fig. 4 and Fig. 7 as follows.
>
> Fig. 4 shows the results of models trained on different *RS* subsets, the numerical accuracies are:
>
> 1. *DINO_Base/16\_Linear (RS0-RS7)*:
>
>       [83.767, 82.65, 83.433, 82.800, 83.467, 83.133, 82.500, 83.033]
>
> 2. *DINO_Small/16\_Linear (RS0-RS4)*:
>
> 	[83.500, 81.950, 83.833, 84.100, 83.553]
>
> 3. *MoCoV2_ResNet50\_Linear (RS0-RS10)*:
>
> 	[71.807, 69.680, 71.987, 71.573, 71.227, 70.487, 71.327, 70.893, 70.353, 70.393, 70.280]
>
> 4. *MAE_Base/16\_Linear (RS0-RS8)*:
>
> 	[77.177, 75.319, 78.215, 77.696, 77.616, 76.098, 75.958, 77.256, 76.558]
>
> 5. *MAE_Base/16\_Finetuning (RS0-RS8)*:
>
> 	[90.855, 89.936, 91.873, 92.212, 91.414, 91.034, 90.855, 90.974, 91.354]
>
> In Fig. 7:
>
>  Configurations/frameworks:
>
> 	 'DINO_Base/16_linear',
> 	 'DINO_Small/16_linear',
> 	 'MoCoV1-ResNet50-200ep',
> 	 'MoCoV2-ResNet50-200ep',
> 	 'MoCoV2-ResNet50-800ep',
> 	 'MAE_Base/16_Linear',
> 	 'MAE_Base/16_Finetuning'
>
> 'RS': 3.0, 2.0, 5.0, 6.0, 7.0, 7.0, 7.0,
>
> 'PD': 4.0, 4.0, 5.0, 6.5, 6.0, 8.0, 7.0,
>
> 'Lottery Ticket': 7.0, 8.0, 7.0, 7.0, 6.0, 7.0, 8.0.
>
> The results are calculated according to the matchness in Table 2. The reason to use plotting as it has better iconicity and vitality which is more visual and intuitive to readers.
>
> We will release all our logs, training code, and pre-trained models for further analysis and reproducibility.
> &nbsp;
> >I'm not convinced by the results that inconsistency is an issue. As shown in Table 2, the Pearson correlation reaches 0.893 using random sampling subsets. After using some simple subset selection procedures, this consistency rises to 0.944.
>
> A: **Inconsistency between the subset and full data** is definitely an issue if we want to use a subset as an indicator for fast hyper-parameter tuning. The Pearson correlation is sensitive to the relative relation of variables between the results trained on the subset and the full data, the improvement here from 0.893 to 0.944 meets the expectation and is not surprising. Moreover, our subset selection policies are not simple, they are carefully thought out and well-designed to defeat the random subset baseline.
> &nbsp;
> >The training accuracy for direct learning on the subsets is missing, as stated in 3.
>
> A: **All the results we provided in the paper are trained directly on the subsets**, including self-supervised learning (Table 2) and supervised learning (Table 3).
> &nbsp;
> >The reproducing details are missing. I can't find the code or the hyperparameters for these experiments.
>
> A: We have provided all the implementation details in Appendix H of the revision. Please check them out. All the code, trained models, and training logs will be made publicly available.
>
> **Please feel free to let us know if you have any further questions.**

---

> ### Author Response · Authors · 2022-11-16
> **Response to Reviewer gNgq (2/3)**
>
> &nbsp;
> >The motivation for choosing a lottery ticket with a consistency principle is ill-justified. For example, the authors stated in 3.2 that " Some trends are inconsistent, for instance, from subset RS-3 to RS-4, the accuracy increases on DINO ViT-Base/16 while decreases on MAE and MoCo" and use this to support their claim that uniform sampling lacks consistency. However, the models are tested via linear probing and the increase in model hyperparameters may result in overfitting or instability during fine-tuning and lead to these performance drops. Overall, this inconsistency may be the result of the difficult balancing of model size and subset size. It does not necessarily reflect the trend of the expressiveness of the models.
>
> A: We clarify **the performing results in this paper are from the models trained on the subsets directly, not the models trained on the full data then crafted.** Further, the hyperparameters we used in our training and testing stages are not increased (not sure what "the increase in model hyperparameters" mean in the comments but we did not adjust the hyperparameters from the original frameworks), they strictly follow the original designs in the individual frameworks (please check out our reversion for all the implementation details.) The mentioned result in Sec. 3.2 is a clear demonstration to prove that the random subsets are not consistent on different configurations/frameworks. Moreover, we clarify the goal of this work is not to reflect the **the trend of the expressiveness of the models**, but the trend of the consistency of the **subset data**.
> &nbsp;
> >I'm doubtful about the training settings. The authors use full-dataset trained backbones to calculate test accuracies and only test if the performance trends are consistent. In most applications, users are more concerned about the performance achieved by directly tuning their model on the subsets. The paper lacks a discussion on this more practical setting. It seems that section 4.2 discusses this subject but only approximate numbers are provided in the radar char and lack precise results.
>
> A: We kindly point out that the understanding is incorrect. In the *Empirical Risk Consistency* selection policy, the full-dataset trained backbones are used to select the target subsets, then we re-train all the models on the subsets to examine the consistency of the performance. We re-iterate **the results in this paper are from the models trained on the subsets, not from the full-dataset trained backbones**.
>
> We have provided the precise/numerical results for the radar chart of section 4.2 in the relevant question below. Please check them out there.

---

> ### Author Response · Authors · 2022-11-16
> **Response to Reviewer gNgq (1/3)**
>
> We thank the reviewer for your time invested in reviewing our paper, and the detailed comments and feedback. Firstly, we would like to clarify our procedure of the proposed framework quickly to avoid any misconception on the procedure of the method, as well as to give a correct understanding. Then, we provide detailed clarifications for each question.
>
> Our setting is indeed what the reviewer cares about, i.e., "users are more concerned about the performance achieved by directly tuning their model on the subsets". We explain it step-by-step as follows:
>
> 1. Previous methods, especially in the self-supervised learning task, usually generate a random subset, and then train different models on this subset to find the optimal configuration, i.e., the hyper-parameter tuning. After that, they transfer the obtained optimal configuration to the full data.
>
> 2. We challenge this practice by verifying whether the random subset is qualified as an indicator to reflect the optimal configuration when training the model on the full data, as done in step 1.
>
> 3. If there is a better subset beyond the random baseline to use as the fast hyper-parameter tuning proxy, we consider it as the dataset lottery ticket.
>
> 4. We try to find out such a subset/subsets by proposing several selection policies. Our baseline policy is the random selection scheme. Specifically,
>
>        Baseline Policy: random selection →  random subsets
>
>        Our Policies: (1) Empirical Risk Consistency; (2) Performance Driven; (3) WordNet Hierarchy; (4) Word Embedding Clustering; (5) Semantic Embedding Clustering; (6) Number Image Reduction → our subsets
>
> 5. Verification: we conduct comprehensive experiments to prove the proposed policy can obtain better subsets than the random baseline and we identify the best one among them.
>
> We hope this quick response can give the reviewer a correct impression of our method procedure. The detailed clarifications are as follows:
> &nbsp;
> >The definition of the data lottery ticket problem is ambiguous. The author stated in Definition 1 that "a subset that has the same or similar empirical behaviors and performance trends as the original full dataset when performing different training approaches and hyperparameters on it".
> It lacks an elaboration on "performance trends" which can be pre-trained performance, training error, generalization error, downstream performance, etc.
>
> A: The "performance trend" indicates that the finetuning/linear probing accuracy of the model trained on the subset data has a consistent trend to the model trained on full data across different configurations. In this work, we care more about the performance trends or relative accuracy trends trained on the subset across different train/eval configurations between the subset and full data. The absolute accuracy on the individual subset is not so necessary for our method.
>
> >The performance trends involve the selection of models. However, different pools of models may have different performance result trends. The author didn't give a clear definition as to how these training structures and training approaches should be chosen.
>
> A: **We only select the subset data from the full dataset, we will not select models**. We follow the same training protocols of the original designs in various frameworks, such as MAE with the ViT\_Base backbone. The implementation details on training and evaluation have been included in Appendix H of the revision. Please check them out there.
> >A formal definition could be given instead of a text description for better clarity.
>
> A: Thanks for the suggestion. We have added a formal definition in section 2 and an explanation in Appendix F.1 for the proposed dataset lottery ticket hypothesis in the revision. Please check them out.

---

### Official Review · Reviewer_xj8W · 2022-10-24

**Confidence:** 3
**Correctness:** 2
**Technical Novelty And Significance:** 3
**Empirical Novelty And Significance:** 2
**Recommendation:** 3

**Clarity, Quality, Novelty And Reproducibility:**

Clarity of the paper needs to be improved before publication. In addition to the points mentioned before:
- The introduction discusses the issue of overfitting while referencing to Figure 1. However, Figure 1 only shows training loss. Additionally, the qualitative difference between the winning ticket and random subsets is not clear from the plots.
- Author should expend the discussion with respect to dataset pruning. Why a dataset pruning method would not allow to find a lottery ticket?
- How is the number of classes selected in the different methods?
- Are the model performances trained on the subset stable for the different classes with respect to the baseline?


**Strength And Weaknesses:**

Strength:
- The paper has sound motivation. SSL research involves a non-trivial computational cost, and it is important to reduce this compute burden to stimulate more research in this area.

Weaknesses:
- It is unclear if the SSL models are retrained on the ‘winning’ subset. If the models are not retrained, this would be a major limitation as model trained on the subset might not have empirical trends similar to the models trained on the full dataset.
- Paper clarity could be improved. I found that the paper was hard to follow. For instance, some important details are missing in the ticket policies description (how do you select the number of classes in PD-{Top,Bottom,…), how do you select the merging order and stopping criteria in WNH-ID. Same question for WEC/SEC. The NIR-ID scheme is not explain in the main text. Figures are also hard to understand. For instance, the x-axis in Figure 6 is not explained. It would be nice to provide more details in the caption and explain the main take-away from the figures.
- If the SSL model are retrained on the smaller subset, what are the computational gain provided by the approach.
-Authors should compare with dataset pruning approaches as they might already find winning lottery tickets.


**Summary Of The Paper:**

The paper investigates the dataset lottery tickets hypothesis in the context of self-supervised learning. The main hypothesis is that one can identify a subset of the data which has empirical trend similar to the full dataset, i.e. hyperparameters that have better performance on the subset also show better performance on the full dataset.

Authors propose several methods to find dataset subsets using a SSL pretrain model and potential label information. Authors empirically investigate which approach can identify winning tickets on the ImageNet dataset.


**Summary Of The Review:**

The paper investigates the question of dataset lottery tickets for self-supervised learning approach. While the topic is interesting and important, the paper clarity needs to be improved.
In particular, it is unclear if the models are re-trained on the dataset subset in the empirical evaluation.
Comparisons with dataset pruning approaches should also be added to better demonstrate the significance of the work.

---

> ### Author Response · Authors · 2022-11-16
> **Response to Reviewer xj8W (2/2)**
>
> &nbsp;
> >Are the model performances trained on the subset stable for the different classes with respect to the baseline?
>
> A: The baseline for our method is the random subset, from our experiments, the selected winning ticket subset shows better stability and robustness than the random subsets. As we mentioned in the introduction section, on each subset and configuration, we train the model three times and report the averaged results, we believe our experimental results are solid for the conclusion. We will make our code, training logs, and trained models publicly available.

---

> ### Author Response · Authors · 2022-11-16
> **Response to Reviewer xj8W (1/2)**
>
> We thank the reviewer for the valuable comments and insightful feedback. Please see our responses and clarifications to the concerns below.
> &nbsp;
> >It is unclear if the SSL models are retrained on the 'winning' subset. If the models are not retrained, this would be a major limitation as model trained on the subset might not have empirical trends similar to the models trained on the full dataset.
>
> A: **All our SSL models are retrained on the random subsets (baselines) and winning ticket subsets**. Our goal is to find out a better subset than the random one in terms of performance consistency by exploring multiple selection policies. The results have demonstrated that the winning ticket subset has a much better consistency than the random subset.
> &nbsp;
> >Paper clarity could be improved. I found that the paper was hard to follow. For instance, some important details are missing in the ticket policies description (how do you select the number of classes in PD-{Top,Bottom,…), how do you select the merging order and stopping criteria in WNH-ID. Same question for WEC/SEC. The NIR-ID scheme is not explain in the main text. Figures are also hard to understand. For instance, the x-axis in Figure 6 is not explained. It would be nice to provide more details in the caption and explain the main take-away from the figures.
>
> A: Thanks for the suggestion. PD-{Top, Bottom, ...} means we use a pre-trained model (e.g., supervised pre-trained EfficientNet-L2) to evaluate samples' accuracy in each class of the ImageNet-1K dataset, then, we rank the accuracy for each class (class-wise) and choose the easiest 100 classes as the PD-Top and the hardest 100 classes as the PD-bottom. We have revised the relevant part of the paper to make it clearer. We will also release the code for selecting subsets to lessen confusion.
>
> The NIR-ID scheme is introduced in Appendix A due to the limits of paper length.
>
> In Fig. 6, the x-axis of model ID from 0-8 indicates that we use different subsets to examine consistency on data augmentation, training budget, and evaluation strategy. The subsets corresponding to each index from 0 to 8 are: CMC subset (random subset); PD-Top; PD-Mid; PD-Bottom; PD-Uniform; RS-1; RS-2; RS-3; RS-4. We have updated the caption in Figure 6 to explain the x-axis corresponding to each model in the revision. Please check it out.
> &nbsp;
> >If the SSL model are retrained on the smaller subset, what are the computational gain provided by the approach. -Authors should compare with dataset pruning approaches as they might already find winning lottery tickets.
>
> A: The state-of-the-art data pruning method [1] can compress the original dataset by ~20%, for comparison, our DLTH can compress by ~90%.
>
> [1] Sorscher, Ben, Robert Geirhos, Shashank Shekhar, Surya Ganguli, and Ari S. Morcos. "Beyond neural scaling laws: beating power law scaling via data pruning." In NeurIPS, 2022.
> &nbsp;
> >The introduction discusses the issue of overfitting while referencing to Figure 1. However, Figure 1 only shows training loss. Additionally, the qualitative difference between the winning ticket and random subsets is not clear from the plots.
>
> A: Fig. 1 demonstrates qualitatively that the winning ticket has a unique training behavior, i.e., training loss, to the random baselines, which is more aligned with the full data. We have updated the relevant part to make this clearer in the revision.
> &nbsp;
> >Author should expend the discussion with respect to dataset pruning. Why a dataset pruning method would not allow to find a lottery ticket?
>
> A: Thanks for the suggestion. We have added the discussions with dataset pruning in Appendix G, as well as in the introduction section. Different from dataset pruning which aims to find a subset that the model trained on it can have similar performance as the model trained on the full data, in our scenario, we do not care about the absolute model’s accuracy trained on the subset but the performance consistency to the models trained on the full data. The two tasks essentially have different goals, and our pruning ratio over the original dataset is also much higher than the dataset pruning method (90% vs. 20%).
> &nbsp;
> >How is the number of classes selected in the different methods?
>
> A: We use 100 classes for *Random Sampling*, *Performance Driven*, *WordNet Hierarchy*, *Word Embedding Clustering*, and *Semantic Embedding Clustering*. For *Empirical Risk Consistency*, there are 367, 155, and 44 classes as shown in Fig. 3's caption according to the performance consistency of pre-trained models. To fairly compare with the random subset baseline, we further reduce 367 classes to 100 classes by random sampling among them, as described in Algorithm 1.

---

### Official Review · Reviewer_kkgk · 2022-10-24

**Confidence:** 2
**Correctness:** 3
**Technical Novelty And Significance:** 3
**Empirical Novelty And Significance:** 3
**Recommendation:** 6

**Clarity, Quality, Novelty And Reproducibility:**

This paper is well-written with reasonable logic. The problem it studies is novel to me. All experiments are repeated multiple times and std-deviations are reported which helps with judging the results. The definition is clear, and the algorithm is easy to follow. Experiments and findings are well presented. Experiments and the conclusions drawn appear to be correct to me – though, I can't say whether this hypothesis holds for other datasets because the authors didn't use more datasets for evaluation.

**Details Of Ethics Concerns:**

No ethics concerns

**Strength And Weaknesses:**

Pros:
1. The paper is well-written and easy to follow.
2. This paper provides a novel perspective on sub-dataset selection.
3. This paper empirically reveals that a randomly selected subset without any prior knowledge is unstable and generally not qualified for reflecting the properties of self-supervised models on the full data. This helps us to use a more reasonable way to screen the important samples of the dataset without misconceptions.

Cons:
1. My major concern is why should we guarantee that the subset categories have consistency in performance similar to that of the full dataset. This is the key contribution/target of this manuscript, and the authors provide an example to explain this issue. However, I am still not sure why we should not select these inconsistent classes in the classification task. For example, can we select some samples which have similar empirical behaviors and performance trends in these classes rather than removing these classes? Dataset Lottery Ticket Hypothesis (DLTH) that recklessly retains all the samples in consistent classes seems to contradict the target of accelerated training.
2. It seems that only a few establishments (e.g., Google, Meta, etc.) can afford the heavy experiments on large-scale datasets training. This issue is caused by large-scale datasets and the over-parameterized networks that go with them. This paper follows a train-select-retrain process, which is hard to reduce the loss of training resources caused by hyper-parameter adjustments (the full dataset has to be fed to the network for training and testing at the beginning). If it is a misunderstanding, please feel free to correct it.
3. This manuscript utilizes empirical risk minimization for evaluating model performances. However, in the case of such a dataset with a large network, serious overfitting often occurs after the dataset is screened. It seems more reasonable to use structure risk minimization to overcome the overfitting problem (e.g., through weight decay) and adjustment of the penalty factor should be a consideration.
4. This manuscript finds the LTH property on datasets, but only verifies this hypothesis on the vision domain, i.e., ImageNet-1K. This is not very convincing although the paper adopts a variety of recent mainstream backbones and baselines. More datasets are appreciated for conducting experiments.


**Summary Of The Paper:**

This paper generalizes Lottery Tickets Hypothesis (LTH) to the subset selection domain, by defining a Dataset Lottery Ticket as a subset that has the same or similar empirical behaviors and performance trends as the original full dataset, which can be identified by some specific approaches (e.g., WordNet Hierarchy). It provides a novel problem that studies the possibility of identifying the subset which can reflect the performance consistency with the full data.

**Summary Of The Review:**

Considering the pros and cons of this paper, I think this paper is good but still has a large space to improve, e.g., more datasets for evaluation.

---

> ### Author Response · Authors · 2022-11-16
> **Response to Reviewer kkgk (2/2)**
>
> &nbsp;
> >This manuscript finds the LTH property on datasets, but only verifies this hypothesis on the vision domain, i.e., ImageNet-1K. This is not very convincing although the paper adopts a variety of recent mainstream backbones and baselines. More datasets are appreciated for conducting experiments.
>
> A: Thanks for the suggestion. We have a quick experimental exploration on AG's News Topic Classification Dataset [1] from NLP domain.
>
> [1] Xiang Zhang, Junbo Zhao, Yann LeCun. Character-level Convolutional Networks for Text Classification. Advances in Neural Information Processing Systems (NIPS 2015).
>
> First, we give a brief introduction of the AG's news topic classification dataset: It is constructed with 4 coarse classes from the original corpus. Each class contains 30,000 training samples and 1,900 testing samples. The total number of training samples is 120,000 and testing 7,600.
>
> Our training/testing code is based on the official PyTorch text classification analysis tutorial using *torchtext library*, our full code for this exploration can be found anonymously at: https://drive.google.com/file/d/1tagJFXlm4o_eiAyylbeqtjQQVlzxns5b/view?usp=sharing.
>
> In our quick experiments, the random (baseline) and lottery ticket subsets are sampled with 10% of the full training data. We examined four configurations as follows:
>
> Configuration_1 (*default*):
>
>     EPOCHS = 10 # epoch
>     LR = 5  # learning rate
>     lr_scheduler = StepLR # lr scheduler
>     BATCH_SIZE = 64 # batch size for training
>
> Configuration_2 (*CosineAnnealingLR*):
>
>     EPOCHS = 10 # epoch;
>     LR = 5  # learning rate;
>     lr_scheduler = CosineAnnealingLR # lr scheduler
>     BATCH_SIZE = 64 # batch size for training
>
> Configuration_3 (*larger bs*):
>
>     EPOCHS = 10 # epoch
>     LR = 5  # learning rate
>     lr_scheduler = StepLR # lr scheduler
>     BATCH_SIZE = 128 # batch size for training
>
> Configuration_4 (*larger lr*):
>
>     EPOCHS = 10 # epoch
>     LR = 10  # learning rate
>     lr_scheduler = StepLR # lr scheduler
>     BATCH_SIZE = 64 # batch size for training
>
> The preliminary results are as follows (C1, C2, C3, C4 are the above four configurations, respectively):
>
> | Train Data   |      C1      |  C2 |       C3      |  C4 | Correlation (ρ$_p$) |
> |----------|:-----------:|------:|:----------:|------:|:---------:|
> | Full | 0.910 |  0.905 |  0.896 | 0.909  |  -- |
> | | | | | | | |
> | RS-0 | 0.864 |  0.862 |  0.862 | 0.863  |  0.764
> | RS-1 | 0.863 |  0.867 |  0.863 | 0.871  | 0.437
> | RS-2 | 0.863 |  0.870 |  0.859 | 0.868  | 0.589
> | RS-3 | 0.870 |  0.868 |  0.863 | 0.864  | 0.617
> | RS-4 | 0.861 |  0.862 |  0.860 | 0.867  | 0.555
> | **Lottery Ticket-0 (ours)** |  0.871 |  0.864 |  0.863 | 0.867  | **0.814**
> | **Lottery Ticket-1 (ours)** |  0.870 |  0.865 |  0.864 | 0.871  | **0.863**
>
> For the lottery ticket subset, we simply select consistent samples across four configurations until reach 10% of the full data. We do this two times to construct different lottery ticket subsets to mitigate variance in different runs. We can observe that the models trained on lottery ticket subsets are much more robust with a higher correlation to the full data.

---

> ### Author Response · Authors · 2022-11-16
> **Response to Reviewer kkgk (1/2)**
>
> We thank the reviewer for the valuable comments and insightful feedback. Please see our responses and clarifications to the concerns below.
> &nbsp;
> >My major concern is why should we guarantee that the subset categories have consistency in performance similar to that of the full dataset. This is the key contribution/target of this manuscript, and the authors provide an example to explain this issue. However, I am still not sure why we should not select these inconsistent classes in the classification task. For example, can we select some samples which have similar empirical behaviors and performance trends in these classes rather than removing these classes? Dataset Lottery Ticket Hypothesis (DLTH) that recklessly retains all the samples in consistent classes seems to contradict the target of accelerated training.
>
> A: We clarify *Empirical Risk Consistency* is one of the selection policies we explored in this paper by guaranteeing that the subset categories have consistency in performance to that of the full dataset. Empirically, we also examined many other policies like *Performance Driven*, *WordNet Hierarchy*, *Semantic Embedding Clustering*, etc., in a comprehensive way. We think these can also be regarded as our contributions to this work.
> &nbsp;
> >It seems that only a few establishments (e.g., Google, Meta, etc.) can afford the heavy experiments on large-scale datasets training. This issue is caused by large-scale datasets and the over-parameterized networks that go with them. This paper follows a train-select-retrain process, which is hard to reduce the loss of training resources caused by hyper-parameter adjustments (the full dataset has to be fed to the network for training and testing at the beginning). If it is a misunderstanding, please feel free to correct it.
>
> A: Thanks for the kind comments. Generally, we have done most of the resource-consuming parts in this work and the follow-up works don't need to repeat them again, such as verifying performance consistency on a large number of various subsets with different selection policies and giving the empirical observations of the current optimal subset (The pre-trained models on the full data with different frameworks can be obtained freely from their official off-the-shelf models). Future studies can focus on exploring better selection policies for choosing better dataset winning tickets. We will release all of our training logs on the various subsets to be beneficial for other research.
> &nbsp;
> >This manuscript utilizes empirical risk minimization for evaluating model performances. However, in the case of such a dataset with a large network, serious overfitting often occurs after the dataset is screened. It seems more reasonable to use structure risk minimization to overcome the overfitting problem (e.g., through weight decay) and adjustment of the penalty factor should be a consideration.
>
> A: Thanks for this suggestion. Structure risk minimization may be an alternative for constructing the subset, while as our goal is to maintain the consistent performance trend between the models trained on the subset and full data, we are not sure whether it can still hold the desired performance consistency. We will leave it as the future work of a new subset selection policy. Actually, we have examined the weight decay factor in Appendix F and our selected subset shows much better robustness than the random subsets. In this work, we have tested multiple selection policies and identified the optimal one empirically, which we believe is a practical and feasible solution for this problem.

---

> ### Comment · Reviewer_kkgk · 2022-12-09
> **Reponse to authors**
>
> Thank you for addressing my questions. I can now better understand this paper. The experimental results from NLP seem promising.
>
> Considering other reviewers' comments and the authors' feedback, I would like to preserve my recommendation (weak acceptance).
>
> --reviewer

---

### Official Review · Reviewer_PcqN · 2022-10-24

**Confidence:** 3
**Clarity, Quality, Novelty And Reproducibility:** None
**Correctness:** 4
**Technical Novelty And Significance:** 2
**Empirical Novelty And Significance:** 3
**Recommendation:** 6

**Strength And Weaknesses:**

### Strength:

1. This paper is well-written and easy to follow.
2. The idea of studying the dataset lottery ticket hypothesis is novel and interesting.
3. Comprehensive experiments on ImageNet-1K across several self-supervised frameworks are conducted, which is a laudable effort.

### Weakness:

I have one concern about this work: for the randomly sampling strategy, what's the performance of randomly sampling partial data points across all classes rather than selecting partial classes, which seems less biased from the full set distribution and serves as a more competitive baseline?

**Summary Of The Paper:**

This work proposes the Dataset Lottery Ticket Hypothesis(DLTH), a novel problem that studies the possibility of identifying the subset which can reflect the performance consistency with the full data. By Empirical Risk Trend, this work demonstrates the existence of dataset-winning tickets. And extensive experiments are conducted across various self-supervised frameworks, which verify the effectiveness and superiority of the proposed dataset-winning ticket policies.

**Summary Of The Review:**

None

---

> ### Author Response · Authors · 2022-11-16
> **Response to Reviewer PcqN**
>
> We thank the reviewer for the supportive comments and feedback. Please see our clarification to your concern below.
> &nbsp;
> >I have one concern about this work: for the randomly sampling strategy, what's the performance of randomly sampling partial data points across all classes rather than selecting partial classes, which seems less biased from the full set distribution and serves as a more competitive baseline?
>
> A: The randomly sampling partial data points across all classes is our NiP selection policy and the results of it are provided in Appendix A. This strategy (NiP) basically will make the subset more diverse but the same category has fewer training samples. In this selection policy, we examined the influence of reducing the number of images in each class for self-supervised learning. To study this, we keep the set of original classes and randomly select 1/10 of the images in each class. It can be observed that the strategy of NiP has low accuracy of 38% on MoCo V2 and 46% on DINO (we use the accuracy from the full data and the low-absolute-performance-subset PD-Bottom as comparisons). Though it is not necessary that the absolute performance is low when the model is trained on the NiP subset since we focus more on the trend across different frameworks, the model trained on such a subset tends to be “dull” and is biased to some particular classes as samples in each class are insufficient compared to the full dataset.

---

### Official Review · Reviewer_drZn · 2022-10-25

**Confidence:** 4
**Clarity, Quality, Novelty And Reproducibility:** The paper is well-written, and the pr…
**Correctness:** 3
**Technical Novelty And Significance:** 2
**Empirical Novelty And Significance:** 2
**Recommendation:** 5

**Strength And Weaknesses:**

Strength:
1. The paper is well motivated, building a performance-consistent sub-training method which can be used as an efficient proxy for hyper-parameter tuning is interesting.

2. The method is simple and easy to implement.

Weakness:

1. Talking about performance consistency, more insightful discussion or theoretical analysis may be required to fully justify the proposed method. Utilizing only 7 models is not enough for this claim.
2. The name “dataset lottery ticket hypothesis” is a little misleading to me. At the first glance, the “dataset lottery ticket hypothesis” may refer to: there exists a subset that can fully retain the model performance.  This paper, however, concentrates on maintaining the relative performance of different models and a few hyper-parameters.

3. Though the proposed ERC strategy is reported can well retain the “performance trends”. Picking samples at class level is quite coarse. There are many other fine-grained data selection methods, ranging from active learning to efficient model training. Though not primarily designed for maintaining “performance trends” consistency, comparing and discussing these methods will strengthen the paper.

4. The rightmost 2 figures of Figure 2 seem to be exactly the same. There are also some typos like subsection 4.1 should refer Figure 6 instead of Figure 7.


minor:
1. It seems that “empirical behaviors and performance trends” refers only to the relative accuracy throughout the paper. I recommend outlining this earlier in the narration.

References:
[1] J T. Ash et.al. Deep Batch Active Learning by Diverse, Uncertain Gradient Lower Bounds. ICLR ‘20

[2] B. Mirzasoleiman et.al. Coresets for Data-efficient Training ofMachine Learning Models. ICML ‘20

[3] G.Citovsky et.al. Batch Active Learning at Scale. NeurIPS ‘21


**Summary Of The Paper:**

This paper inspects the dataset lottery ticket hypothesis, where training on just subsets have similar empirical behaviors and performance trends as training on the full set. So that analysis and hyper-parameter tuning can be conducted efficiently. Various sampling strategy with different models are compared, the proposed ERC strategy achieves the best consistency.

**Summary Of The Review:**

Interesting paper that tries to build a performance-consistent sub-training method. However, there are some issues mentioned in the weakness part. I would like to raise my score if these questions are properly addressed.

---

> ### Author Response · Authors · 2022-11-16
> **Response to Reviewer drZn**
>
> We thank the reviewer for the insightful comments and valuable suggestions. Please see our responses and clarifications to the concerns below.
> &nbsp;
> > Talking about performance consistency, more insightful discussion or theoretical analysis may be required to fully justify the proposed method. Utilizing only 7 models is not enough for this claim.
>
> A: Thanks for this suggestion. We have added brief discussions from the theoretical perspective through a teacher-student perceptron scheme to explain our DLTH following [1] in the revision of Appendix F.1. Beyond 7 models, we actually have also examined the factors of batch size and weight decay effects on the performance consistency in Appendix F. We have revised this part in the revision to make it clearer.
>
> [1] Sorscher, Ben, Robert Geirhos, Shashank Shekhar, Surya Ganguli, and Ari S. Morcos. "Beyond neural scaling laws: beating power law scaling via data pruning." In NeurIPS, 2022.
> &nbsp;
> > The name “dataset lottery ticket hypothesis” is a little misleading to me. At the first glance, the “dataset lottery ticket hypothesis” may refer to: there exists a subset that can fully retain the model performance. This paper, however, concentrates on maintaining the relative performance of different models and a few hyper-parameters.
>
> A: Thanks for pointing this out, we consider slightly modifying the title to “Does Dataset Lottery Ticket Hypothesis Exist on Performance Consistency?” to avoid this misleading impression. Please kindly help us confirm this.
> &nbsp;
> > Though the proposed ERC strategy is reported can well retain the “performance trends”. Picking samples at class level is quite coarse. There are many other fine-grained data selection methods, ranging from active learning to efficient model training. Though not primarily designed for maintaining “performance trends” consistency, comparing and discussing these methods will strengthen the paper.
>
> A: Thanks for the suggestion. The discussion and comparison with active learning are provided in Appendix G due to the limits of the main paper length. We have added part of them in the introduction section of the revision to make the difference between active learning and our method clearer. For the fine-grained selection methods, we actually have done an exploration of sample-wise selection within each class in Appendix A of the original submission.
> &nbsp;
> > The rightmost 2 figures of Figure 2 seem to be exactly the same. There are also some typos like subsection 4.1 should refer Figure 6 instead of Figure 7.
>
> A: Yes, here we use the same figure in Fig. 2 as an illustration to show the motivation of trend consistency between full data and the winning ticket subset. We will replace it if it is necessary. For the reference issue of Figure 6 instead of Figure 7 in Sec. 4.1, we have corrected it in the revision.
>
> minor:
> &nbsp;
> > It seems that “empirical behaviors and performance trends” refers only to the relative accuracy throughout the paper. I recommend outlining this earlier in the narration.
>
> >References:
>
> >[1] J T. Ash et.al. Deep Batch Active Learning by Diverse, Uncertain Gradient Lower Bounds. ICLR ‘20
>
> >[2] B. Mirzasoleiman et.al. Coresets for Data-efficient Training ofMachine Learning Models. ICML ‘20
>
> >[3] G.Citovsky et.al. Batch Active Learning at Scale. NeurIPS ‘21
>
> A: Thanks for this suggestion. Yes, the “empirical behaviors and performance trends” refers only to the relative accuracy in this work. We have highlighted this in the abstract section using a footnote in the revision.
>
> We have also accommodated the background section in Appendix G to discuss more on active learning and other relevant topics, and all the mentioned references have been included in our revision.

---

> > ### Comment · Reviewer_drZn · 2022-12-11
> > **Thanks for the response**
> >
> > Thanks for the response, I went through Appendix F.1, but the theoretical analysis seems just a replicate of the analysis in (Sorscher et al., 2022). Adding more explanation of how this generalization analysis incorporates performance consistency will further improve the paper I think.

---

> > > ### Author Response · Authors · 2022-12-11
> > > **Thanks for your further comments**
> > >
> > > We appreciate your additional comments. Our analysis is not a repetition of Sorscher et al., 2022. Data pruning (Sorscher et al., 2022) aims to make the students trained on the pruned dataset have the same or similar decision boundaries as the teachers which are trained on the original full data, i.e., the same or close test error ε between teachers and students, so that the student models can be as accurate as the teacher model, and the selected data can be the desired pruned dataset.
> > >
> > > Here, we solely utilize the teacher-student paradigm to explain our selection mechanism but the criteria are completely different. Instead of letting the decision boundaries of the students be the same as the teachers, it’s also difficult in our scenario since our subset has only 10% of the full data, while data pruning (Sorscher et al., 2022) retains 80% from the full data. We concentrate on the consistency of test error ε which is calculated using the same way in Sorscher et al., 2022 for the final perceptron as a function across different configurations of teachers and students. The consistency metrics we used are *Pearson Correlation Coefficient* and  *Spearman’s Rank Correlation* which have been introduced in the paper.
> > >
> > > In summary, Sorscher et al., 2022’s criterion and analysis for sub-data selection are to find the same or close decision boundary between teacher and student through the same or close test error ε. However, as we described in the revision, we focus on the consistency of test error ε trend for the final perceptron as a function. Thus, the two analyses are basically different. We hope our clarifications address your concern.

---

### Author Response · Authors · 2022-11-06
**Response to the reviewers**

Dear Reviewers,

We appreciate your time and efforts invested in the reviewing. Your comments are undoubtedly constructive and valuable, which will definitely help us improve the quality of this paper. We will provide a revision to accommodate all of your suggestions, such as more results on other datasets, experiments with more configurations, etc., shortly.

However, we also find that the value of this study may be somewhat underestimated by a few reviewers.

We would like to clarify quickly the value of DLTH and the differences/advantages over dataset pruning methods:

&nbsp;
>1. The value of this work [for Reviewer gNgq]:

A: The reviewer mentioned that the consistency principle is not well justified and could be much influenced by subset size and model structures. This is true and it indeed reflects the value of this work. Our goal is to raise attention that the random subset has this issue with monitoring the performance of the full dataset across different architectures that is congenitally imprecise, while this practice has been widely used for this purpose. Our method cannot completely solve the mismatch between full data and subset, but considering that this study is the preliminary trial, and the provided lottery ticket subset is much more robust on multiple/diverse configurations than the randomly selected subset, it can be a significant progress on this problem.

&nbsp;
>2. Our dataset lottery ticket hypothesis is different from **dataset pruning** on at least two aspects which we have expressed in the paper [for Reviewer xj8W]:

(i) Pruning ratio. The state-of-the-art data pruning method can only compress ~20% of the dataset [1], while our DLTH can compress ~90%, which is more efficient to use.

[1] Ben Sorscher, Robert Geirhos, Shashank Shekhar, Surya Ganguli, and Ari S Morcos. Beyond neural scaling laws: beating power law scaling via data pruning. arXiv preprint arXiv:2206.14486, 2022.

(ii) Dataset pruning will change the optimal configuration and hyper-parameter choices since it changes the distribution of the original dataset. However, the goal of DLTH is to find a subset that can reflect the same or similar configuration behavior on the full dataset, which is more practical if people stick to the original dataset and would like to find an optimal configuration for the original full dataset, rather than the pruned replacement. The criterion of dataset pruning is the performance on the remaining data, while the criterion for our DLTH is the **performance trend** instead of the absolute accuracy. They are quite different from each other.

&nbsp;
>3. The potential applications of the proposed DLTH:

A: Many works in neural architecture search (NAS) adopt performance prediction to assist the searching process. DLTH is naturally applicable for this, which, besides hyper-parameter tuning, can be very useful in the related research of fast performance prediction.

We hope these quick clarifications can give reviewers an overview and a picture of our main contributions in this work. As we have emphasized above (most of the reviewers also mentioned that this work is interesting, novel from the sub-dataset perspective, and important), without this work, it seems people still did not realize that using a random subset as a performance indicator in self-supervised learning for algorithm development is dangerous. Of course, the current shape of the proposed method is not perfect for monitoring the identical behavior of full data by subsets, we reiterate the provided lottery ticket subset has been much more robust than the widely-used random one and this is an important progress in this direction. Our detailed responses to questions from each reviewer will be provided very soon, as well as a revised version of the paper. Thanks again to all reviewers.

---

### Author Response · Authors · 2022-11-17
**Reviewer comments incorporated in newest revision**

Thanks again to all the reviewers for the helpful suggestions! We've uploaded a new version of the paper that incorporates the reviewer comments. These include:

1. A clearer clarification of the dataset lottery ticket hypothesis and a brief explanation from the theoretical perspective.
2. More details of training and testing settings.
3. Results on a new dataset (AG's News Topic Classification Dataset) from the NLP domain.
4. Polishing other minor typos, as well as the presentation of the paper.

----------------------------------------------------------------------------------------------------

[**For Reviewer gNgq**] We also provide condensed clarifications here for your comments in case our formal response is too heavy.

**1. The training accuracy for direct learning on the subsets is missing.**

All our results provided in the paper are learned/retrained directly on the subset, not on the full data then recombine.

**2. The performance trends involve the selection of models.**

The models are trained using their default settings of the frameworks. We will not select models but *select proper subset data*.

**3. I'm not convinced by the results that inconsistency is an issue.**

Inconsistency on performance between subset and full data is the key issue we aim to address. Only when it is solved, we then can leverage a subset as the accuracy indicator for fast hyperparameter tuning/estimation tasks.

**4. It does not necessarily reflect the trend of the expressiveness of the models.**

We DO NOT care about the  *expressiveness of the models* but the *performance consistency* of the subset and full data.

**5. The class semantic-based methods for choosing lottery tickets are hard to apply beyond labeled image datasets.**

This semantic clustering policy is designed for the scenario that the human-annotated labels are not available, since we adopt a DINO pre-trained model as the feature extractor and an unsupervised clustering method is employed.

**6. The authors use full-dataset trained backbones to calculate test accuracies and only test if the performance trends are consistent.**

  As Reviewer kkgk summarized in the comments, our method follows a train-select-retrain process, not "only test if the performance trends are consistent".

----------------------------------------------------------------------------------------------------

Finally, we put a considerable amount of time and effort into the rebuttal and revised version of the paper because we recognize the importance of the work for the community. We hope the reviewer will read our response, if you have any further questions, they are welcome to raise and we would be pleased to respond. We will make our code, models, and logs publicly available so that other research can benefit from them.

---

### Author Response · Authors · 2022-12-06
**We are looking forward to your post-rebuttal feedback**

Dear All Reviewers,

We hope things are going well. We’d like to thank you again for taking the time to review our paper, and we have responded to all of your questions one by one. Given that the deadline for the discussion is drawing near, please let us know if our clarifications have addressed your concerns, or if you still have questions.

Best,

Paper1098 Authors

---

### Decision · Program_Chairs · 2023-01-20

**Decision:**

Reject

**Justification For Why Not Higher Score:**

The paper isn't ready to be published yet. The writing/framing are rough and confusing, and the scientific setup/evaluation aren't solid yet.

**Justification For Why Not Lower Score:**

N/A

**Metareview: Summary, Strengths And Weaknesses:**

*Summary:* State-of-the-art deep learning settings are expensive to run. So expensive, in many cases, that they are not suitable for doing preparatory science; only the final hero run. It's typical for researchers to try out and refine a method in small-scale settings (e.g., MNIST or CIFAR) before testing it in large-scale settings. This paper is interested in the question of which small-scale settings provide meaningful signal for large-scale settings, specifically in the context of self-supervised learning (SSL) in computer vision (CV). It focuses on removing classes from a dataset, which seems to be a common way of scaling down in SSL for CV. The paper proposes a heuristic (Empirical Risk Trend) that seems to outperform other metrics (including randomly dropping classes, which is common in the literature) according to the criteria specified in the paper. This criterion is about "consistency": are the relative orders of performance of the models preserved? This concept of consistency proved tricky.

*Strengths:*
* The paper asks an important question. It's a question which, as soon as it has a convincing scientific answer, would be of great value to the community. It would put state-of-the-art research within reach for many more researchers than is currently the case.
* The authors ran a lot of experiments; they poured a lot of resources into getting to an answer on this question (which is unfortunately necessary). To understand the relationship between small-scale and large-scale training, you have to train a lot of small-scale and large-scale models.
* The paper takes a reasonable first crack at a very thorny question. (See weaknesses for more on that thorniness.)

*Weaknesses:* The reviewers struggled over several key foundational concepts in the paper, so I read the paper in-depth myself to make sure I understood where the points of disagreement were.
* Framing and branding. The paper obscures a wonderfully simple set of ideas in a tangle of jargon that makes it very hard to suss out what's really going on. I think this is where much of the disagreement between reviewers and authors came from: it was exceedingly tricky to get to the core idea. The "data lottery ticket hypothesis" branding isn't really on target for what this paper does, and it seemed only to confuse reviewers by invoking the wrong intuitions. Yes, you can shoehorn this paper into a lottery-ticket-like framework if you want to. No, it didn't seem helpful to do so. The notion of "consistency" was also poorly defined (even after revisions), and it led to an enormous amount of reviewer confusion. To the authors: the reviewers are responsible for digging into the paper in-depth whether they like it or not; if they found it this confusing, it needs revision so that casual readers don't get driven away immediately.
* The reviewers struggled with both understanding the definition of "consistency" and accepting that it was the right thing to measure in the paper. I did too. It took me (and them) a while to wrap our heads around the fact that the goal wasn't to get models with the same absolute performance (this is where the lottery ticket metaphor was especially detrimental), and then there was debate over whether this was the right definition. The reviewers didn't seem convinced, and I wasn't either:
    * Is the way consistency was instantiated the right one? Why does trend matter, and not just the relative ordering?
    * Are the ways of evaluating sufficient? Why was the only comparison across different models/training paradigms, and not across something like hyperparameter search?
    * In both cases, I recommend that the authors reframe the paper to be driven by a couple of specific use-cases, such as comparing across different architectures/training paradigms (already in the paper) and other important tasks someone might perform that require similar comparisons (hyperparameter search for (e.g.,) learning rate, searching for the number of epochs across which to train). Not only would having clearer motivating scenarios be more convincing to reviewers, but it would also provide justification for why this particular way of measuring consistency makes sense (and not something different, more general, or more specific). Personally, I'm not convinced that this measure of consistency isn't too specific (why do the actual performance numbers matter and not just the ordering, for example?).

I think this paper has the beginnings of a really useful paper (good question, a good first stab at answering it), but it's not ready for publication in terms of the quality of the science and the quality of the writing. The reviewers seem to generally agree on that.

**One last note to the authors:** The reviewers were confused about a lot of fundamental points in the paper's setup. This isn't on the reviewers. The framing is genuinely confusing. I ignored places where they were off base, but the fact that they were confused so badly is a problem in itself.

**Summary Of Ac-Reviewer Meeting:**

N/A